# Safe and Sparse Newton Method for Entropic-Regularized Optimal Transport

**Zihao Tang**
School of Statistics and Data Science
Shanghai University of Finance and Economics
`tangzihao@stu.sufe.edu.cn`

**Yixuan Qiu**[*]
School of Statistics and Data Science
Shanghai University of Finance and Economics
`qiuyixuan@sufe.edu.cn`

## Abstract

Computational optimal transport (OT) has received massive interests in the machine learning community, and great advances have been gained in the direction of entropic-regularized OT. The Sinkhorn algorithm, as well as its many improved versions, has become the *de facto* solution to large-scale OT problems. However, most of the existing methods behave like first-order methods, which typically require a large number of iterations to converge. More recently, Newton-type methods using sparsified Hessian matrices have demonstrated promising results on OT computation, but there still remain a lot of unresolved open questions. In this article, we make major new progresses towards this direction: first, we propose a novel Hessian sparsification scheme that promises a strict control of the approximation error; second, based on this sparsification scheme, we develop a *safe* Newton-type method that is guaranteed to avoid singularity in computing the search directions; third, the developed algorithm has a clear implementation for practical use, avoiding most hyperparameter tuning; and remarkably, we provide rigorous global and local convergence analysis of the proposed algorithm, which is lacking in the prior literature. Various numerical experiments are conducted to demonstrate the effectiveness of the proposed algorithm in solving large-scale OT problems.

## 1 Introduction

In recent years, optimal transport (OT, [36]) has received massive attentions from the deep learning community, and has become a fundamental modeling tool in modern statistical machine learning [35, 23]. One major challenge of applying OT to real-life problems, the computation of large-scale OT, has also gained many progresses in the direction of approximate OT methods, especially the entropic-regularized OT [6].

Consider two discrete probability measures $\mu = \sum_{i=1}^{n} a_i \delta_{x_i}$ and $\nu = \sum_{j=1}^{m} b_j \delta_{y_j}$, where $a = (a_1, \ldots, a_n)^T$ and $b = (b_1, \ldots, b_m)^T$ are two vectors satisfying $\sum_{i=1}^{n} a_i = \sum_{j=1}^{m} b_j = 1$, $a_i > 0$, $b_j > 0$, $i = 1, \ldots, n$, $j = 1, \ldots, m$, $\{x_i\}_{i=1}^{n}$ and $\{y_j\}_{j=1}^{m}$ are two sets of data points, and $\delta_x$ is the Dirac measure at position $x$. Without loss of generality we assume that $n \geq m$, as their roles can be exchanged. Let $M = (M_{ij}) \in \mathbb{R}^{n \times m}$ be a cost matrix, whose entry $M_{ij}$ typically represents the

---

[*]Corresponding author.

38th Conference on Neural Information Processing Systems (NeurIPS 2024).

distance between data points $x_i$ and $y_j$. Also define $\Pi(a,b) = \{T \in \mathbb{R}^{n \times m} : T\mathbf{1}_m = a, T^T\mathbf{1}_n = b, T \geq 0\}$, where the inequality sign applies elementwisely. Then OT between the two measures $\mu$ and $\nu$ can be characterized by the following linear programming problem,

$$\min_{P \in \Pi(a,b)} \langle P, M \rangle, \tag{1}$$

where $\langle A, B \rangle = \mathrm{tr}(A^T B)$. Assuming $n = m$, the computational complexity of solving (1) using standard linear programming solvers is typically at the order of $O(n^3 \log(n))$ [29], which can be difficult even for moderate $n$ and $m$. One approach to approximating the solution is to add an entropic penalty term to the objective function, leading to the entropic-regularized OT problem [6]:

$$\min_{T \in \Pi(a,b)} \langle T, M \rangle - \eta h(T), \tag{2}$$

where $h(T) = \sum_{i=1}^{n} \sum_{j=1}^{m} T_{ij}(1 - \log T_{ij})$ is the entropy term, and $\eta > 0$ is a regularization parameter. The objective function in (2) is $\eta$-strongly convex on $\Pi(a,b)$, so (2) has a unique global solution, denoted as $T^*$. The matrix $T^*$ is often referred to as the Sinkhorn transport plan. Problem (2) can be solved by the celebrated Sinkhorn algorithm [38, 33], which is based on efficient matrix-vector multiplication operations. Due to its computational advantage, the Sinkhorn algorithm has become the *de facto* solution to large-scale OT problems for a long time.

However, from a practical point of view, the Sinkhorn algorithm typically demonstrates a sub-linear convergence speed, thus requiring a large number of iterations for a moderately high precision. More recently, second-order methods, such as the Newton method, have attracted a growing number of researchers to rethink the solution to (2) [3, 34]. It has been shown that (2) has a dual problem (Proposition 4.4 of [30]):

$$\max_{\alpha, \beta} \mathcal{L}(\alpha, \beta) := \max_{\alpha, \beta} \alpha^T a + \beta^T b - \eta \sum_{i=1}^{n} \sum_{j=1}^{m} e^{\eta^{-1}(\alpha_i + \beta_j - M_{ij})}, \quad \alpha \in \mathbb{R}^n, \beta \in \mathbb{R}^m. \tag{3}$$

Let $\alpha^* = (\alpha_1^*, \ldots, \alpha_n^*)^T$ and $\beta^* = (\beta_1^*, \ldots, \beta_m^*)^T$ be one optimal solution to (3), and then the Sinkhorn transport plan $T^*$ can be recovered as $T^* = \tau(\alpha^*, \beta^*)$, where for two vectors $\alpha = (\alpha_1, \ldots, \alpha_n)^T$ and $\beta = (\beta_1, \ldots, \beta_m)^T$, $\tau(\alpha, \beta)$ is a matrix with entries

$$[\tau(\alpha, \beta)]_{ij} = \exp\left\{\eta^{-1}(\alpha_i + \beta_j - M_{ij})\right\}.$$

Remarkably, (3) is equivalent to a smooth and unconstrained convex optimization problem, so it can be solved using first-order methods such as gradient descent, or second-order methods including the Newton method. It is well known that under some smoothness assumptions on the objective function, the Newton method has a fast quadratic local convergence rate, and hence it typically requires much fewer iterations than the Sinkhorn algorithm. However, each iteration in the Newton method involves solving a dense linear system, resulting in a per-iteration cost of $O(n^3)$, which is nearly of the same order as that of an unregularized OT problem. To this end, [34] proposes the Sinkhorn-Newton-Sparse (SNS) algorithm, which approximates the Hessian matrix of (3) by a sparse one, so that each iteration roughly has a computational cost of $O(n^2)$, the same order as the Sinkhorn algorithm.

The SNS algorithm demonstrates promising results for solving the entropic-regularized OT, yet there remain a number of open questions. First, the SNS algorithm relies on initial Sinkhorn iterations to achieve a moderately small optimality gap, but in practice, the cost for this warm initialization also needs to be taken into account. Second, though SNS advocates using the conjugate gradient (CG) method to solve the sparse linear systems, there is no strong guarantee on the invertibility of the sparsified Hessian matrix. Third, [34] conjectures that SNS has a super-linear local convergence rate, but it is not yet proven by theoretical analysis.

In this article, we make major progresses on the second-order method for entropic-regularized OT by resolving the three challenges above. First, we carefully design a new sparsification algorithm that has a well-controlled approximation error, and then propose a *safe* Newton-type algorithm, in the sense that the linear systems for computing the search directions are always positive definite. This property addresses the invertibility issues of existing sparsified Newton methods, and is crucial for practical implementation. We also show that the proposed algorithm has a global convergence guarantee, so no initial Sinkhorn iterations are needed. Most importantly, we provide solid theoretical analysis of the proposed method, and show that it achieves a quadratic local convergence rate similar to the vanilla Newton method.

**Contribution**   Our main contribution compared to prior art is summarized as follows:

1. We propose a novel Hessian sparsification scheme that promises a strict control of the approximation error.
2. Based on this sparsification scheme, we prove that the sparsified Hessian matrix is always positive definite regardless of the sparsification parameter. This property then leads to a *safe* Newton-type method that is guaranteed to avoid singularity in computing the search directions.
3. The developed algorithm has a clear implementation for practical use, avoiding most hyperparameter tuning. An efficient implementation is included in the **RegOT** Python package[2].
4. We provide rigorous global and local convergence analysis of the proposed algorithm, which is lacking in prior literature.

## 2   Background and Related Work

**Notation**   Throughout this article, we use $f(x)$, $g(x)$, and $H(x)$ to represent the objective function, gradient, and Hessian matrix at point $x$, respectively. For a matrix $A$, let $A_{i\cdot}$ be the vector of the $i$-th row of $A$, and $A_{\cdot j}$ be the vector of the $j$-th column of $A$. For a vector $v = (v_1, \ldots, v_n)^T$, let $\tilde{v}$ denote the vector $(v_1, \ldots, v_{n-1})^T$, and for a matrix $A = (A_{\cdot 1}, \ldots, A_{\cdot n})$, let $\tilde{A}$ represent the matrix $(A_{\cdot 1}, \ldots, A_{\cdot,n-1})$. The symbol $\|\cdot\|$ stands for the Euclidean norm for a vector argument, and represents the operator norm for matrices. The $\ell_1$-norm and infinity norm are denote by $\|\cdot\|_1$ and $\|\cdot\|_\infty$, respectively, applying to both vectors and matrices.

**The main objective**   As has been explained in Section 1, the key to the computation of entropic-regularized OT is to solve its smooth dual problem (3), which is equivalent to the convex optimization problem $\min_{\alpha,\beta} -\mathcal{L}(\alpha, \beta)$. However, it is worth noting that the variables $(\alpha, \beta)$ have one redundant degree of freedom: if $(\alpha^*, \beta^*)$ is one solution to (3), then so is $(\alpha^* + c\mathbf{1}_n, \beta^* - c\mathbf{1}_m)$ for any $c$. Therefore, to ensure identifiability, we globally set $\beta_m = 0$ without loss of generality. In what follows, we use the vector $x = (\alpha^T, \tilde{\beta}^T)^T \in \mathbb{R}^{n+m-1}$ to collect all free dual variables. Clearly, we always have $\beta = (\tilde{\beta}^T, \beta_m)^T = (\tilde{\beta}^T, 0)^T$, so $\tilde{\beta}$ and $\beta$ will be used interchangeably when we consider functions of $\beta$. Then the main target of this article is to efficiently solve the optimization problem

$$\min_{x \in \mathbb{R}^{n+m-1}} f(x) := -\mathcal{L}(\alpha, \beta) = \eta \mathbf{1}_n^T \tau(\alpha, \beta) \mathbf{1}_m - \alpha^T a - \beta^T b. \tag{4}$$

It is known that $f(x)$ is a strictly convex function, so if (4) has a solution, then it is unique. It has also been proven in the existing literature that the gradient and Hessian matrix of $f(x)$ have the following closed-form expressions (see for example [21]):

$$g(x) = -\begin{bmatrix} \nabla_\alpha \mathcal{L}(\alpha, \beta) \\ \nabla_{\tilde{\beta}} \mathcal{L}(\alpha, \beta) \end{bmatrix} = \begin{bmatrix} T\mathbf{1}_m - a \\ \tilde{T}^T \mathbf{1}_n - \tilde{b} \end{bmatrix}, \qquad T = \tau(\alpha, \beta), \tag{5}$$

$$H(x) = -\begin{bmatrix} \nabla_\alpha^2 \mathcal{L}(\alpha, \beta) & \nabla_{\tilde{\beta}}\left(\nabla_\alpha \mathcal{L}(\alpha, \beta)\right) \\ \left[\nabla_{\tilde{\beta}}\left(\nabla_\alpha \mathcal{L}(\alpha, \beta)\right)\right]^T & \nabla_{\tilde{\beta}}^2 \mathcal{L}(\alpha, \beta) \end{bmatrix} = \eta^{-1} \begin{bmatrix} \mathbf{diag}(T\mathbf{1}_m) & \tilde{T} \\ \tilde{T}^T & \mathbf{diag}(\tilde{T}^T \mathbf{1}_n) \end{bmatrix}.$$

**Computational OT**   There are a huge number of methods developed to solve (2) and (4). The Sinkhorn algorithm [38, 33, 6] can be interpreted as applying the block coordinate descent (BCD), also known as the alternating minimization (AM) method, to (4). Along this direction, many extensions of the Sinkhorn algorithm have been proposed. For example, [13] develops an accelerated version of AM, and a greedy variant of the Sinkhorn algorithm, named Greenkhorn, is developed in [1]. Accelerated first-order methods, such as the adaptive primal-dual accelerated gradient descent (APDAGD, [10]) and adaptive primal-dual accelerated mirror descent (APDAMD, [19]), have been proposed to solve the constrained problem (2). Moreover, since the dual problem (4) is smooth, various quasi-Newton methods, such as the limited-memory Broyden–Fletcher–Goldfarb–Shanno (L-BFGS) method [20], have also been used [7]. Second-order methods are less common in the literature to solve (4), mainly due to the high cost of solving the linear systems, and [34] proposes a practical sparsified Newton method to approximate the true Hessian by a sparse matrix. Another direction to utilize the sparsity is the importance sparsification method [18], which replaces the dense kernel matrix in the Sinkhorn algorithm by a sparsified one through random sampling.

---

[2] https://github.com/yixuan/regot-python.

**Newton-type methods** The Newton method is a classical second-order optimization algorithm that has been extensively studied. Assuming $f$ is twice continuously differentiable, the Newton method generates a sequence of iterates $\{x_k\}$ using the updating formula $x_{k+1} = x_k - [H(x_k)]^{-1}g(x_k)$. However, computing the linear system $[H(x_k)]^{-1}g(x_k)$ with a dense Hessian matrix generally costs $O(n^3)$ of computation, so the sparsification of $H(x_k)$ with a solid convergence guarantee is the main focus of this article. The Newton method also has many extensions, for example the regularized Newton method to solve problems with non-isolated solutions [17], and the inexact Newton method [8] that allows inexact solutions to the linear system $[H(x_k)]^{-1}g(x_k)$. Our proposed method use a combination of these ideas to reduce the computational cost of the classical Newton method.

**OT in machine learning** OT is a blooming research topic in modern machine learning research. As a powerful tool to characterize the transformation of statistical distributions, OT has wide applications in deep generative models [2, 11, 14], domain adaptation and transfer learning [5, 4], and fairness in machine learning [12, 25, 32], among many others. Readers are referred to review articles such as [35, 23] for a summary of machine learning tasks and methods that utilize OT.

## 3 The Proposed Algorithm

To complement existing second-order methods for entropic-regularized OT, in this article we propose the SSNS algorithm, short for safe and sparse Newton method for Sinkhorn-type OT. It has two central components: a new scheme to sparsify the Hessian matrix, and a Newton-type algorithm to update iterates.

### 3.1 Sparsifying the Hessian matrix

Ideally, the sparsified Hessian matrix, denoted by $H_\delta(x)$, should meet two criteria: first, it should be close to the true $H(x)$ with a tunable approximation error; second, it needs to preserve the positive definiteness of the original Hessian matrix. The first point is to ensure that $H_\delta(x)$ does not lose too much information of the true $H(x)$. The second point is especially important in implementing the Newton method, since a linear system $[H_\delta(x)]d = -g(x)$ needs to be solved to compute the search direction $d$, and one needs to make sure that $H_\delta(x)$ is invertible. These two points are briefly mentioned in [34] as an implementation practice, but without a strong theoretical guarantee.

To this end, we introduce a new adaptive method to sparsify $H(x)$, which takes the two points above as *first principles* in both theoretical analysis and practical implementation. To describe the whole algorithm, first define an operator `select_small`$(v, \delta)$, which takes a vector $v = (v_1, \ldots, v_n)^T$ and a scalar $\delta > 0$ as inputs, and outputs a mask vector $\phi = (\phi_1, \ldots, \phi_n)^T$ in the following way. Suppose that $v_{\pi(1)} \le v_{\pi(2)} \le \cdots \le v_{\pi(n)}$ are the sorted values of $v$, where $\pi(s)$ is the original index of the $s$-th smallest element in $v$. We also define $\pi^{-1}(\cdot)$ to be the inverse of $\pi(\cdot)$, *i.e.*, $\pi^{-1}(i)$ is the integer such that $\pi(\pi^{-1}(i)) = i$. Let $S$ be the largest integer such that $\sum_{s=1}^{S} v_{\pi(s)} \le \delta$. Then we set $\phi_i = 1$ if $\pi^{-1}(i) \le S$, and $\phi_i = 0$ otherwise. For example, if $v = (2, 1, 3, 5, 2)^T$ and $\delta = 6$, then we have $\phi = (1, 1, 0, 0, 1)^T$, since $1 + 2 + 2 \le \delta$ but $1 + 2 + 2 + 3 > \delta$.

Next, define an operator `apply_mask`$(v, \phi)$, which inputs a vector $v = (v_1, \ldots, v_n)^T$ and a mask $\phi = (\phi_1, \ldots, \phi_n)^T$, and outputs a vector $u = (u_1, \ldots, u_n)^T$ with $u_i = \phi_i \cdot v_i$. For the example above, we have `apply_mask`$(v, \phi) = (2, 1, 0, 0, 2)^T$. Then given the current dual variables $x = (\alpha^T, \tilde{\beta}^T)^T$, Algorithm 1 outputs a sparse matrix $H_\delta$ as an approximation to the true Hessian matrix $H(x)$.

It is worth noting that in the construction of $H_\delta$, the diagonal elements $\mathbf{diag}(T\mathbf{1}_m)$ and $\mathbf{diag}(\tilde{T}^T\mathbf{1}_n)$ use the *original* $T$ matrix, whereas the off-diagonal elements are the sparsified $T_\delta$ matrix, with the last column removed. In this way, Theorem 2 shows that $H_\delta$ indeed has a well-controlled approximation error as requested by our first criterion.

**Theorem 1** (Approximation error of sparsification). *Let $H_\delta$ be output by Algorithm 1 with a given vector $x$, and define $D = H(x) - H_\delta$. Then for any $\delta \ge 0$, we have $D \ge 0$, where the inequality sign holds elementwise, and for each $i, j = 1, 2, \ldots, n + m - 1$,*

$$\|D_{i\cdot}\|_1 \le \eta^{-1}\delta, \quad \|D_{\cdot j}\|_1 \le \eta^{-1}\delta.$$

*Moreover, $\|D\| \le \eta^{-1}\delta$.*

---

**Algorithm 1** Sparsifying the Hessian matrix.

---

**Input:** Dual variable vector $x = (\alpha^T, \tilde{\beta}^T)^T$, threshold parameter $\delta \geq 0$
**Output:** Sparsified Hessian matrix $H_\delta$
 1: Initialize a zero matrix $\Delta \in \mathbb{R}^{n \times m}$ and compute $T = \tau(\alpha, \beta)$
 2: **for** $j = 1, 2, \ldots, m-1$ **do**
 3: $\quad \phi \leftarrow \texttt{select\_small}(T._j, \delta), \quad \Delta._j \leftarrow \texttt{apply\_mask}(T._j, \phi)$
 4: **for** $i = 1, 2, \ldots, n$ **do**
 5: $\quad \phi \leftarrow \texttt{select\_small}(\Delta_{i\cdot}, \delta), \quad \Delta_{i\cdot} \leftarrow \texttt{apply\_mask}(\Delta_{i\cdot}, \phi)$
 6: $T_\delta \leftarrow T - \Delta$
 7: $H_\delta \leftarrow \eta^{-1} \begin{bmatrix} \mathbf{diag}(T\mathbf{1}_m) & \tilde{T}_\delta \\ \tilde{T}_\delta^T & \mathbf{diag}(\tilde{T}^T\mathbf{1}_n) \end{bmatrix}$

---

Theorem 1 shows that all the row-wise and column-wise $\ell_1$-norms of the residual matrix $D$ is bounded by the pre-specified threshold $\delta$, up to a constant multiplier $\eta^{-1}$. More importantly, the overall operator norm of $D$ also has the same upper bound, and this property plays a key role in analyzing the convergence behavior of the SSNS algorithm later in Section 3.2.

Next, Theorem 2 validates our second criterion on $H_\delta$: the sparsified Hessian matrix by Algorithm 1 is guaranteed to be positive definite, *regardless of* the threshold parameter $\delta$. Therefore, it can be *safely* used to compute the Newton search directions.

**Theorem 2** (Positive definiteness). *Suppose that $n \geq m$. For any $\alpha \in \mathbb{R}^n$, $\tilde{\beta} \in \mathbb{R}^{m-1}$, and $M \in \mathbb{R}^{n \times m}$, let $T = \tau(\alpha, \beta)$, $r = T\mathbf{1}_m$, and $c = T^T\mathbf{1}_n$. Then for any $\delta \geq 0$, the matrix $H_\delta$ output by Algorithm 1 is always positive definite. Specifically, we have*

$$\sigma_{\max}(H_\delta) \leq 2\eta^{-1} \cdot \max\{\|r\|_\infty, \|c\|_\infty\}, \qquad \sigma_{\min}(H_\delta) \geq \eta^{-1} \cdot \frac{n-m+1}{2n} \cdot \min_{i,j} T_{ij},$$

*where $\sigma_{\max}(\cdot)$ and $\sigma_{\min}(\cdot)$ represent the largest and smallest eigenvalues of a symmetric matrix, respectively.*

The two features presented by Theorem 1 and Theorem 2 are crucial in designing the optimization algorithm and analyzing its theoretical properties, which we introduce in details in Section 3.2.

### 3.2 The SSNS algorithm

In Theorem 2, we have shown that the sparsified Hessian matrix $H_\delta$ is always positive definite, thus invertible. However, in actual computation, $H_\delta$ may be nearly singular, thus causing numerical instability. To further stabilize the optimization procedure, we propose a safe and sparse Newton method inspired by the regularized Newton method [17] and the Levenberg–Marquardt algorithm [16, 22].

One of the key ingredients of SSNS is to introduce a shift parameter $\lambda$ in solving the Newton linear system, leading to a search direction of the form $p = -(H_\delta + \lambda I)^{-1}g(x)$ in each iteration. The shift parameter $\lambda$ has multiple functions. First, it stabilizes the vanilla Newton linear system $H_\delta^{-1}g(x)$, as the matrix $H_\delta + \lambda I$ has a smaller condition number than $H_\delta$, which potentially accelerates iterative linear solves such as the CG method. Second, if we allow $\lambda$ to vary along iterations, $\lambda$ has the effect of adjusting the search direction when $-H_\delta^{-1}g(x)$ is not a good candidate. Third, as we will later show in the convergence analysis, an adaptive $\lambda$ plays an important role in both the global and local convergence of the algorithm.

On the other side, the sparsification threshold $\delta$ is also essential to the design of the algorithm, as we need to control the approximation error of $H_\delta$ during the iterations. Intuitively, when the current iterate $x_k$ is close to the optimum, $H_\delta$ should be also close to the true Hessian matrix $H(x_k)$ to achieve the fast convergence rate, whereas when $x_k$ is far away, it may be beneficial to use a large $\delta$, since $H_\delta$ is then more sparse and leads to a faster computation of the linear system $-(H_\delta + \lambda I)^{-1}g(x)$.

Overall, in SSNS we allow $\delta$ and $\lambda$ to vary in each iteration, and use the notation $\delta_k$ and $\lambda_k$ to reflect this adaptivity. Intuitively, both $\delta_k$ and $\lambda_k$ cause approximation errors to the true Hessian matrix in determining the Newton search direction, so they should tend to zero when $x_k$ is sufficiently close to

the optimum. On the other hand, we do not want $\delta_k$ and $\lambda_k$ to decay too fast, since otherwise they make $H_\delta$ too dense and the linear system may be ill-conditioned. To this end, we set their values according to the current gradient norm $\|g(x_k)\|$, which can be viewed as an indicator of the distance between $x_k$ and the optimum $x^*$. Specifically, we set $\lambda_k = \mu_k\|g(x_k)\|$ and $\delta_k = \nu_0\|g_k\|^\gamma$, where $\mu_k$ is a parameter that dynamically changes in each iteration, and $\nu_0 > 0$ and $\gamma \geq 1$ are two constants. These settings are justified by the convergence analysis later in Theorems 3 and 5.

Before presenting the full details, we first introduce a few functions and quantities that play important roles in the algorithm. In each iteration, SSNS computes a quantity $\rho_k$,

$$\rho_k = \frac{f(x_k) - f(x_k + \xi_k p_k)}{m_k(0) - m_k(\xi_k p_k)},$$

where $\xi_k > 0$ is a step size parameter, and the $m_k(\cdot)$ function is a local quadratic approximation to the objective function:

$$m_k(p) = f(x_k) + [g(x_k)]^T p + \frac{1}{2} p^T H_{\delta_k} p. \tag{6}$$

The quantity $\rho_k$ has two functions: first, it adjusts the $\mu_k$ parameter, and second, it also determines whether one should accept the proposed new iterate $x_k + \xi_k p_k$. A negative value of $\rho_k$ means that the objective function actually increases at the new iterate, so one should reject the proposed point, and increase $\lambda_k$ in the next iteration for a potentially better search direction. The overall SSNS algorithm is then given in Algorithm 2.

---

**Algorithm 2** Safe and sparse Newton method for Sinkhorn-type optimal transport.

---

**Input:** Initial point $x_0$, parameters $\{\mu_0, \nu_0, c_l, c_u, \kappa\} > 0$, $\gamma \geq 1$, $\rho_0 \in (0, \frac{1}{2})$, $\varepsilon_{tol} > 0$
**Default values:** $\mu_0 = 1$, $\nu_0 = 0.01$, $c_l = 0.1$, $c_u = 1$, $\kappa = 0.001$, $\gamma = 1$, $\rho_0 = \frac{1}{4}$
**Output:** $x_k$
 1: **for** $k = 0, 1, 2, \ldots$ **do**
 2:     Compute $g_k = g(x_k)$, $\delta_k = \nu_0\|g_k\|^\gamma$
 3:     **if** $\|g_k\| < \varepsilon_{tol}$ **then**
 4:         **return** $x_k$
 5:     Compute $H_{\delta_k}$ according to Algorithm 1 with $x \leftarrow x_k$
 6:     Compute $p_k = -(H_{\delta_k} + \mu_k\|g_k\|I)^{-1} g_k$
 7:     Select any $\xi_k \in [c_l, c_u]$
 8:     Compute $\rho_k = \dfrac{f(x_k) - f(x_k + \xi_k p_k)}{m_k(0) - m_k(\xi_k p_k)}$, $m_k(\cdot)$ is defined in (6)
 9:     Update $\mu_{k+1} = \begin{cases} 4\mu_k, & \text{if } \rho_k < \rho_0 \\ \max\{\mu_k/2, \kappa\}, & \text{if } \rho_k \geq 1 - \rho_0 \\ \mu_k, & \text{otherwise} \end{cases}$
10:     **if** $\rho_k > 0$ **then**
11:         $x_{k+1} = x_k + \xi_k p_k$
12:     **else**
13:         $x_{k+1} = x_k$

---

### 3.3 Step size selection

It is worth noting that in Algorithm 2, the step size $\xi_k$ can be taken an *arbitrary* value from a fixed interval $[c_l, c_u]$. This is quite different from typical line search methods, which require the new iterate $x_k + \xi_k p_k$ to satisfy certain conditions, such as the sufficient decrease condition and the curvature condition [24]. In other words, it is one of the advantages of SSNS that the cost of step size selection is predictable: we can consider a *fixed* number of candidates, and select one of them according to some criterion. At one extreme, it is completely acceptable to always set $\xi_k = 1$. However, in practice, we advocate Algorithm 3 to heuristically choose $\xi_k$, which considers a fixed number of candidates, and computes the objective function value for each candidate. The algorithm will early stop if a decrease in function value is found. If all candidates increase the function value, then return the step that results in the smallest function value. Based on our empirical results, setting $(\xi_{[0]}, \xi_{[1]}, \xi_{[2]}, \xi_{[3]}) = (1, 0.5, 0.25, 0.1)$ gives reasonably good performance in most numerical experiments.

**Algorithm 3** A practical step size selection method for Algorithm 2.

---

**Input:** Candidate step sizes $1 = \xi_{[0]} > \xi_{[1]} > \cdots \xi_{[N]} > 0$, current $x_k$, $p_k$, objective function $f(\cdot)$
 1: Initialize $\xi^* = 1$, $f^* = +\infty$
 2: **for** $i = 0, 1, \ldots, N$ **do**
 3:     Compute $x_{\text{trial}} = x_k + \xi_{[i]} p_k$, $f_{\text{trial}} = f(x_{\text{trial}})$
 4:     **if** $f_{\text{trial}} < f^*$ **then**
 5:         $\xi^* = \xi_{[i]}$, $f^* = f_{\text{trial}}$
 6:     **if** $f^* < f(x_k)$ **then**
 7:         **return** $\xi^*$, $f^*$
 8: **return** $\xi^*$, $f^*$

---

## 4 Theoretical Guarantees on Convergence

In this section, we present our major theoretical contributions to the sparsified Newton method for entropic-regularized OT. We first show that starting from an arbitrary initial point $x_0$, the iterates generated by Algorithm 2 eventually converge to the unique global optimum $x^*$. Therefore, SSNS does not require a warm initialization using the Sinkhorn algorithm as in [34].

**Theorem 3** (Global convergence guarantee). *Let $\{x_k\}$ be generated by Algorithm 2, and $x^*$ is an optimal point of (4). Then either Algorithm 2 terminates in finite iterations, or $x_k$ satisfies*

$$\lim_{k \to \infty} \|g(x_k)\| = 0, \quad \lim_{k \to \infty} \|x_k - x^*\| = 0.$$

The next theorem characterizes the behavior of SSNS when the iterates are sufficiently close to the optimum. Since in Algorithm 2, the proposed new iterate $x_k + \xi_k p_k$ is rejected if $\rho_k \leq 0$, it is important to show that such rejections will be very rare as the algorithm proceeds.

**Theorem 4.** *There exists an integer $K > 0$ such that for all $k \geq K$, $\rho_k \geq 1 - \rho_0$, $\mu_{k+1} \leq \kappa$, and $x_{k+1} = x_k + \xi_k p_k$.*

Theorem 4 indicates that eventually the parameter $\mu_k$ is upper bounded by the pre-specified value $\kappa$, making the shift parameter of the Newton method, namely, $\mu_k \|g_k\|$, tend to zero. This is consistent with the intuition we have described in Section 3.2. More importantly, the quantity $\rho_k$ will also be greater than the threshold $1 - \rho_0$, so the new step proposed by the Newton direction will always be accepted, regardless of the bounds $c_l$ and $c_u$. This makes SSNS quite flexible in picking the step sizes, unlike classical line search algorithms that pose various conditions that need to be satisfied.

Finally, Theorem 5 shows that SSNS indeed has a quadratic local convergence rate that matches the Newton method based on genuine and dense Hessian matrices.

**Theorem 5** (Quadratic local convergence rate). *Fix $\xi_k \equiv 1$. Then there exists an integer $K' > 0$ and a constant $L > 0$ such that for all $k \geq K'$,*

$$\|x_{k+1} - x^*\| \leq L \|x_k - x^*\|^2.$$

Overall, Theorems 3 to 5 provide solid theoretical guarantees on the proposed SSNS algorithm, which are a major complement to the existing literature on solving entropic-regularized OT using realistic second-order methods. More importantly, they validate that SSNS is indeed a "safe" algorithm that is quite robust to the choice of initial value, step sizes, strength of sparsification, and hyperparameters.

## 5 Numerical Experiments

In this section, we test the performance of the proposed SSNS algorithm on various numerical experiments[3]. There are a huge number of algorithms developed for entropic-regularized OT, and we focus on representative ones from each category of optimization methods: 1. the Sinkhorn algorithm as the default option for entropic-regularized OT, interpreted as a BCD method; 2. the APDAGD

---

[3]The programming code to reproduce the results is available at `https://github.com/TangZihao1997/SSNS`.

algorithm for accelerated first-order method; 3. the L-BFGS algorithm for quasi-Newton method; 4. the globalized Newton method with line search; 5. the proposed SSNS algorithm.

We first consider three benchmark datasets to define the OT problem, each with two ways of constructing the cost matrices. More explanations of the experiment setting are given in Section A.2.

- **(Fashion-)MNIST**: OT between a pair of images from the MNIST [15] or Fashion-MNIST [37] dataset. The $a$ and $b$ vectors are flattened and normalized pixel values, and the cost matrix holds the $\ell_1$-distances or squared Euclidean distances between individual pixels. The problem size is $n = m = 784$.
- **ImageNet**: OT between two categories of images from the ImageNet dataset [9]. We use a subset of ImageNet from the Imagenette Github repository[4], which contains ten classes of ImageNet images. Approximately 1000 images per category are selected. We map each image to a 30-dimensional feature vector by first passing the image to a ResNet18 network, resulting in a 512-dimensional vector, then followed by a dimension reduction by principal component analysis. Let $x_i \in \mathbb{R}^{30}$ be the feature vector of an image in the first category, $i = 1, \ldots, n$, and $y_j \in \mathbb{R}^{30}$ be the feature vector of an image in the second category, $j = 1, \ldots, m$. Then $a = n^{-1}\mathbf{1}_n$, $b = m^{-1}\mathbf{1}_m$, and the cost matrix is $M_{ij} = \|x_i - y_j\|_1$ or $M_{ij} = \|x_i - y_j\|^2$. The problem size is $n \approx m \approx 1000$.

To make the regularization parameter $\eta$ comparable in different settings, we normalize all cost matrices to have unit infinity norms, namely, $M \leftarrow M/\|M\|_\infty$. Then we consider two settings of the regularization parameter, $\eta = 0.01$ and $\eta = 0.001$.

For entropic-regularized OT, a commonly-used criterion to evaluate optimality is the marginal error of the estimated transport plan. Let $x_k = (\alpha_k^T, \tilde{\beta}_k^T)^T$ be the current iterate for dual variables, and set $T_k = \tau(\alpha_k, \beta_k)$. Then the marginal error is given by $\sqrt{\|T_k\mathbf{1}_m - a\|^2 + \|T_k^T\mathbf{1}_n - b\|^2}$. Coincidentally, this is exactly the current gradient norm $\|g(x_k)\|$ as indicated by (5). Figure 1 shows the plots of marginal errors against iteration number or run time for different algorithms on the three benchmark datasets.

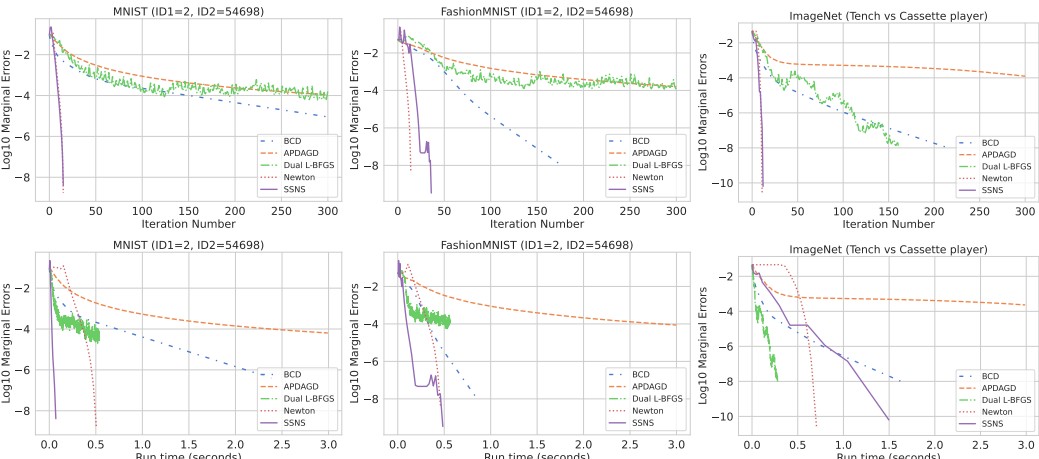

Figure 1: Top: Marginal error vs. iteration number for different algorithms on three datasets. Bottom: Marginal error vs. run time. The cost matrix is based on the $\ell_1$-distance, and $\eta = 0.01$.

From the top row of Figure 1, it is clear that second-order methods have much faster convergence speed compared to first-order and quasi-Newton methods. However, for the vanilla Newton method, this advantage is weakened by its high per-iteration cost, resulting in less competitive run time performance as shown in the bottom row of Figure 1. The SSNS algorithm avoids this issue by using sparse matrix operations, and hence for the run time results, it still shows an order of magnitude speedup for MNIST and Fashion-MNIST data. The run time for SSNS is relatively longer in the ImageNet dataset, mostly because the transport plan is more dense under $\eta = 0.01$. We show in

---
[4]https://github.com/fastai/imagenette.

Figure 2 that as $\eta$ becomes smaller, the Hessian matrix can be better approximated by a sparse matrix, thus enlarging the advantage of SSNS. This point is consistent with Theorem 1 of [34], which states that smaller $\eta$ in general results in better sparse approximation. More discussions on the impact of regularization parameter is given in Section A.3.

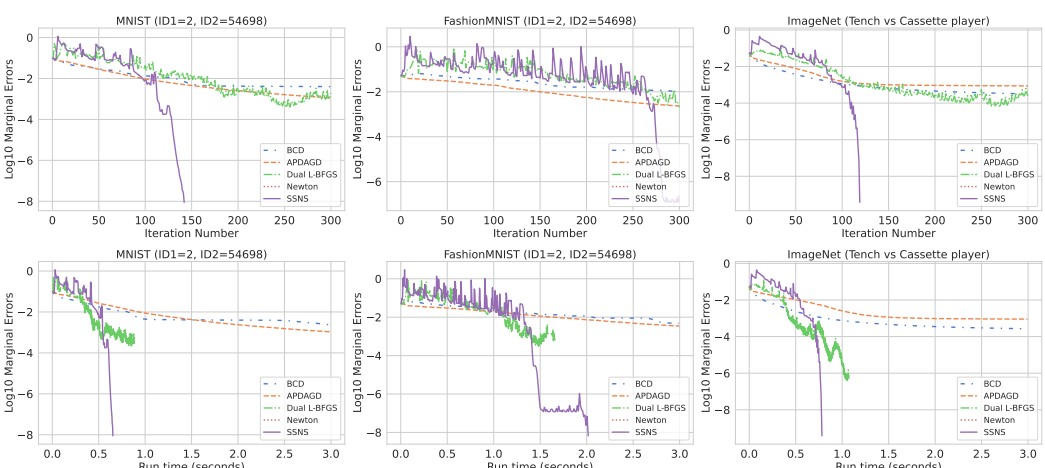

Figure 2: Top: Marginal error vs. iteration number for different algorithms on three datasets. Bottom: Marginal error vs. run time. The cost matrix is based on the $\ell_1$-distance, and $\eta = 0.001$.

We also note that with $\eta = 0.001$, the Newton method encounters numerical issues in solving the linear systems: the Hessian matrix is nearly singular, and the line search procedure fails along the computed search direction. Therefore, the Newton method does not generate any valid results in Figure 2. This issue further validates our intuition in Section 3.2: when $H(x)$ is nearly singular, $-[H(x)]^{-1}g(x)$ may not be a good search direction, and the introduction of the shift parameter $\lambda$ in SSNS may give a better candidate $-(H_\delta + \lambda I)^{-1}g(x)$.

In the ImageNet experiments above, we map each image to a 30-dimensional feature vector to compute the cost matrix. To study the impact of the feature dimension, in Section A.4, we conduct additional experiments with $d = 60, 90, 200, 300, 500$, under the same setting as in Figure 2.

Next, we show in Figure 3 the results for cost matrices based on squared Euclidean distances. The implications are similar: the Newton method fails in the ImageNet dataset, and costs too much run time in the other two datasets. In contrast, SSNS shows great computational advantage. The experiment results for more test cases are given in Section A.5 in the appendix.

Finally, to study the scalability of SSNS, we consider the following synthetic OT problem that can generate data with arbitrary dimensions.

- **Large-scale synthetic data**: The basic setting is to approximate the OT between two continuous distributions: the source is an exponential distribution with mean one, and the target is a normal mixture distribution $0.2 \cdot N(1, 0.2) + 0.8 \cdot N(3, 0.5)$. We discretize the problem in the following way: let $x_i = 5(i-1)/(n-1)$, $i = 1, \ldots, n$, and $y_j = 5(j-1)/(m-1)$, $j = 1, \ldots, m$, which are equally-spaced points on [0, 5]. Define the cost matrix as $M_{ij} = (x_i - y_j)^2$. Let $f_1$ and $f_2$ be the density functions of the source and target distributions, respectively. Then we set $\tilde{a}_i = f_1(x_i)$, $\tilde{b}_j = f_2(y_j)$, $a_i = \tilde{a}_i/\left(\sum_{k=1}^n \tilde{a}_k\right)$, and $b_j = \tilde{b}_j/\left(\sum_{k=1}^m \tilde{b}_k\right)$.

Similar to the previous experiment setting, we normalize the cost matrix, $M \leftarrow M/\|M\|_\infty$, and set $\eta = 0.001$. We then solve the problem at the scales of $n = m = 1000, 5000$, and $10000$, but only considering BCD and SSNS, as other methods are too time-consuming to proceed. The results are visualized in Figure 4, whose pattern is clear: BCD demonstrates a linear-like convergence rate, and SSNS has a fast convergence speed consistent with the theoretical quadratic rate. Thanks to the Hessian sparsification, SSNS does not suffer from a high per-iteration cost, so overall it provides an efficient solver for entropic-regularized OT even on very large problems.

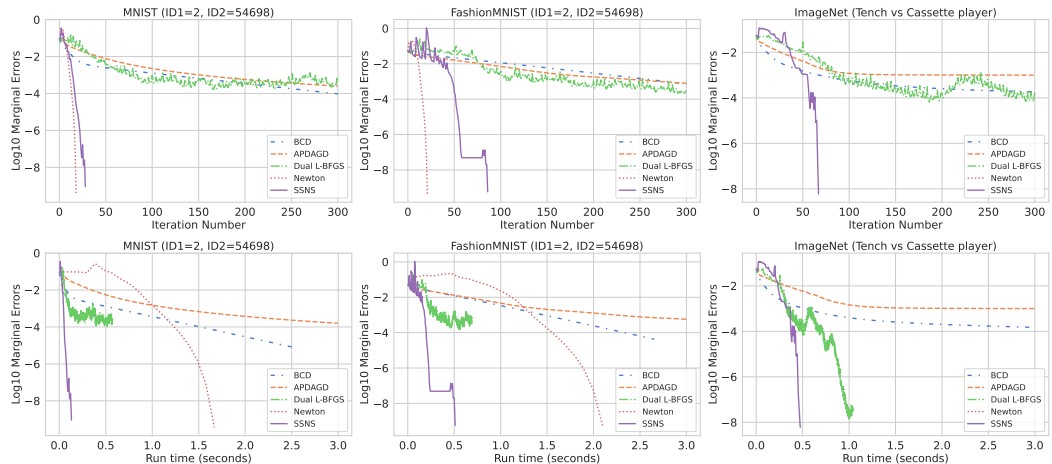

Figure 3: Top: Marginal error vs. iteration number for different algorithms on three datasets. Bottom: Marginal error vs. run time. The cost matrix is based on the squared Euclidean distance, and $\eta = 0.001$.

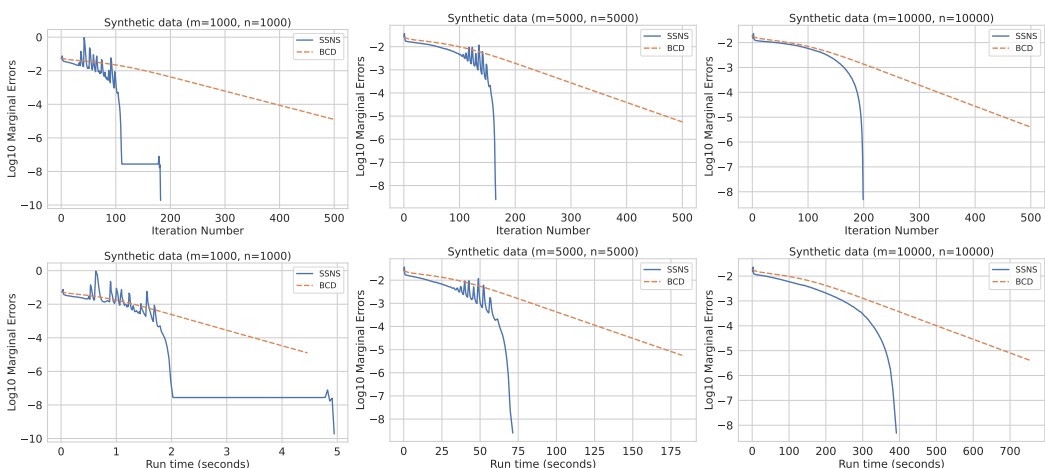

Figure 4: Comparing BCD and SSNS on large OT problems. Top: Marginal error vs. iteration number for different problem sizes. Bottom: Marginal error vs. run time.

## 6   Conclusion

In this article, we have carefully designed and analyzed a new variant of the sparsified Newton method for solving large-scale entropic-regularized OT problems. As demonstrated by [34], the sparsified Newton method typically enjoys visibly faster convergence speed than first-order methods, and meanwhile maintains a comparable computational cost per iteration. This paper substantially complements the existing literature on sparsified Newton method on the following aspects. First, a new Hessian sparsification method is developed, which nicely interacts with the convergence theory. Second, based on this scheme, we prove that the sparsified Hessian matrix is always positive definite, thus promising a safe Newton direction computation. Third, we have a thorough theoretical analysis of the proposed SSNS algorithm, which affirmatively answers a previous conjecture made in [34] on the convergence speed. Finally, our algorithm features an easy and clear implementation that does not require warm initialization or heavy hyperparameter tuning, thus being user-friendly for practical use.

**Limitations**   A potential limitation of the proposed method is that for some specific OT problems, the transport plan may not be well approximated by a sparse matrix. In such cases, the Hessian matrix may be too dense even after the adaptive sparsification step.

## Acknowledgements

Yixuan Qiu's work was supported in part by National Natural Science Foundation of China (12101389), Shanghai Pujiang Program (21PJC056), MOE Project of Key Research Institute of Humanities and Social Sciences (22JJD110001), and Shanghai Research Center for Data Science and Decision Technology.

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

# A   Additional Experiment Details

## A.1   Computing environment

All experiments in this article are conducted on a personal computer with an Intel i9-13900K CPU, 32 GB memory, and a Ubuntu 24.10 operating system.

## A.2   Motivation

The experiments in Section 5 are designed to reflect two typical uses of OT. In the (Fashion-)MNIST experiment, images are vectorized as density vectors, and OT is a used as a tool for image morphing and interpolation [27]. We follow the experiment settings in the existing literature such as [28] and [34] to define $a$, $b$, and $M$: given two images, let $a \in \mathbb{R}^{784}$ be the vectorized and normalized pixel values of the first image, and similarly define $b \in \mathbb{R}^{784}$ for the second image. For pixel $i$ in the first image and pixel $j$ in the second image, let $(w, h)$ and $(w', h')$ be their original coordinates in the image, respectively. Then the cost value between pixel $i$ and pixel $j$ is

$$M_{ij} = |w - w'| + |h - h'|$$

based on the $\ell_1$-distance, or

$$M_{ij} = (w - w')^2 + (h - h')^2$$

base on the squared Euclidean distance. The whole cost matrix $M \in \mathbb{R}^{784 \times 784}$ then collects all pairwise cost values $M_{ij}$.

The ImageNet experiment uses OT as a statistical distance to measure the difference between two distributions. In this setting, each image is one observation of a distribution, and we use OT to compute the (approximate) Wasserstein distance between two classes of images. Since each image is mapped to a 30-dimensional feature vector, the entries of the cost matrix are simply the $\ell_1$-distances or squared Euclidean distances between the feature vectors of two images.

## A.3   Impact of the regularization parameter

In Figures 1 and 2, we have shown that the regularization parameter $\eta$ may affect the performance of optimization algorithms. To further study this effect, we consider a series of equally-spaced $\eta$ values in the logarithmic domain, and display the run times and number of iterations of each algorithm. For brevity, we focus on the comparison between BCD and SSNS, and the results are given in Table 1.

Table 1: Performance comparison between BCD and SSNS under different regularization parameters for the ImageNet experiment in Section 5. The convergence tolerance is set to $\varepsilon_{tol} = 10^{-8}$. Left: cost matrix based on the $\ell_1$-distance. Right: cost matrix based on the squared Euclidean distance.

| $\log_{10}(\eta)$ | Method | Time (s) | Iterations | $\log_{10}(\eta)$ | Method | Time (s) | Iterations |
|---|---|---|---|---|---|---|---|
| -2 | BCD | 1.628 | 217 | -2 | BCD | 0.438 | 59 |
|  | SSNS | 1.523 | 13 |  | SSNS | 2.235 | 11 |
| -2.25 | BCD | > 3.765 | > 500 | -2.25 | BCD | 0.853 | 114 |
|  | SSNS | 0.960 | 20 |  | SSNS | 1.066 | 15 |
| -2.5 | BCD | > 3.765 | > 500 | -2.5 | BCD | 3.327 | 443 |
|  | SSNS | 0.461 | 30 |  | SSNS | 0.997 | 23 |
| -2.75 | BCD | > 3.766 | > 500 | -2.75 | BCD | > 3.773 | > 500 |
|  | SSNS | 0.383 | 57 |  | SSNS | 0.529 | 35 |
| -3 | BCD | > 3.767 | > 500 | -3 | BCD | > 3.773 | > 500 |
|  | SSNS | 0.771 | 120 |  | SSNS | 0.458 | 68 |

From Table 1 we can find that BCD is very sensitive to the value of $\eta$. When $\eta$ is large, BCD may demonstrate some computational advantages, but when $\eta$ is small, BCD typically fails to meet the error tolerance within 500 iterations. The pattern of SSNS shows some interesting points: when $\eta$ becomes smaller, the number of iterations also increases, but the overall runtime of SSNS may even

decrease. This is because smaller $\eta$ values typically result in more sparse Hessian approximations, thus leading to faster sparse linear system solving. These findings are consistent with our explanations in Section 5.

## A.4 Impact of the feature dimension

In the ImageNet experiment in Section 5, we map each image to a 30-dimensional feature vector to compute the cost matrix. To study the impact of the feature dimension, we conduct five more experiments with $d = 60, 90, 200, 300, 500$, under the same setting as in Figure 2. The results are shown in Figure 5. The plots demonstrate similar patterns, implying that the convergence property of SSNS is robust to the feature dimension of input images.

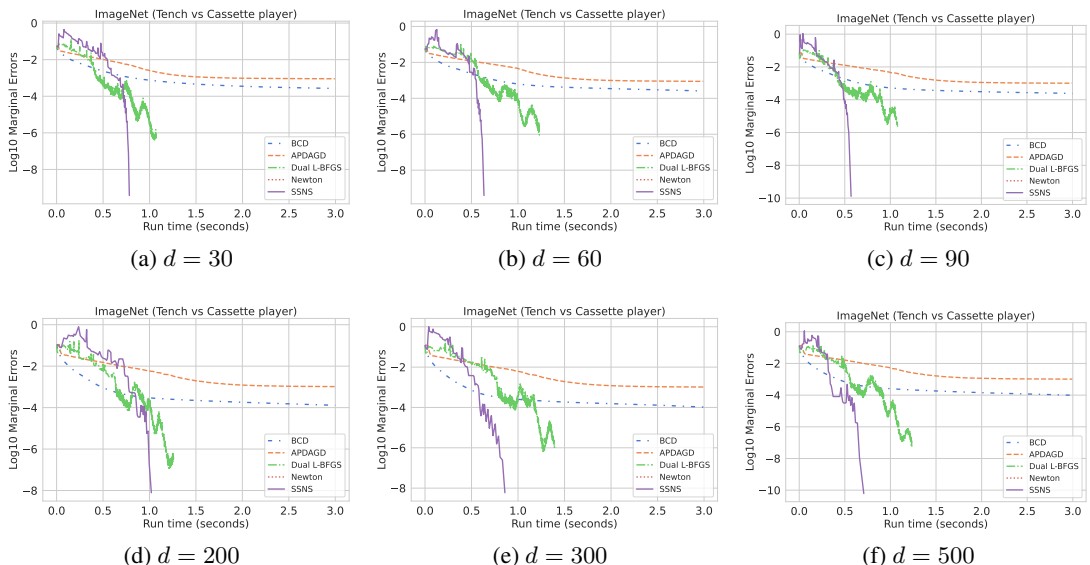

Figure 5: ImageNet experiment with different feature dimensions. The cost matrix is based on the $\ell_1$-distance, and $\eta = 0.001$.

## A.5 Additional test cases

Besides the numerical experiments in Section 5, here we include two more test cases from each dataset. The first test case shown in Figure 6 uses $\ell_1$-distances to construct cost matrix, and the second one, demonstrated in Figure 7, is based on squared Euclidean distances. Both test cases use the regularization parameter $\eta = 0.001$.

# B Proof of Theorems

## B.1 Proof of Theorem (1)

In this proof all inequality signs hold elementwisely when applied to vectors and matrices.

By Algorithm (1), $T_\delta = T - \Delta$, where each element $\Delta_{ij}$ of $\Delta$ is either zero, or the corresponding element $T_{ij}$ of $T$. Clearly, we have $\Delta \geq 0$. Due to the selection scheme in Algorithm (1), each row sum and each column sum of $\Delta$ is upper bounded by the threshold $\delta$, $i.e.$, $\Delta \mathbf{1}_m \leq \delta \mathbf{1}_n$ and $\Delta^T \mathbf{1}_n \leq \delta \mathbf{1}_m$. Since the last column of $\Delta$ is always a zero vector by the design of the algorithm, we also easily get $\tilde{\Delta} \mathbf{1}_{m-1} = \Delta \mathbf{1}_m \leq \delta \mathbf{1}_n$ and $\tilde{\Delta}^T \mathbf{1}_n = \tilde{d} \leq \delta \mathbf{1}_{m-1}$, where $d = \Delta^T \mathbf{1}_n$.

Note that

$$D = H(x) - H_\delta = \eta^{-1} \begin{bmatrix} O & \tilde{\Delta} \\ \tilde{\Delta}^T & O \end{bmatrix} := \begin{bmatrix} D_1 \\ D_2 \end{bmatrix},$$

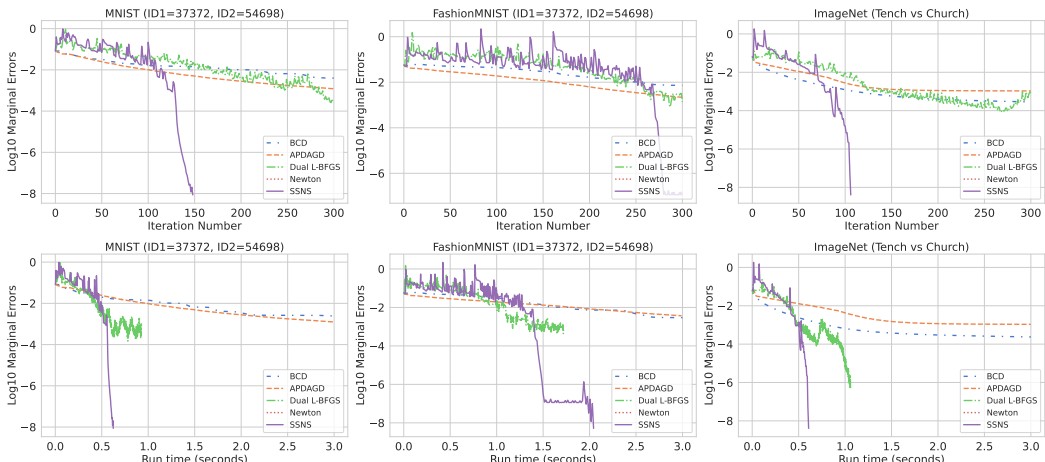

Figure 6: Additional test cases based on $\ell_1$-distances and $\eta = 0.001$. Top: Marginal error vs. iteration number for different algorithms on three datasets. Bottom: Marginal error vs. run time.

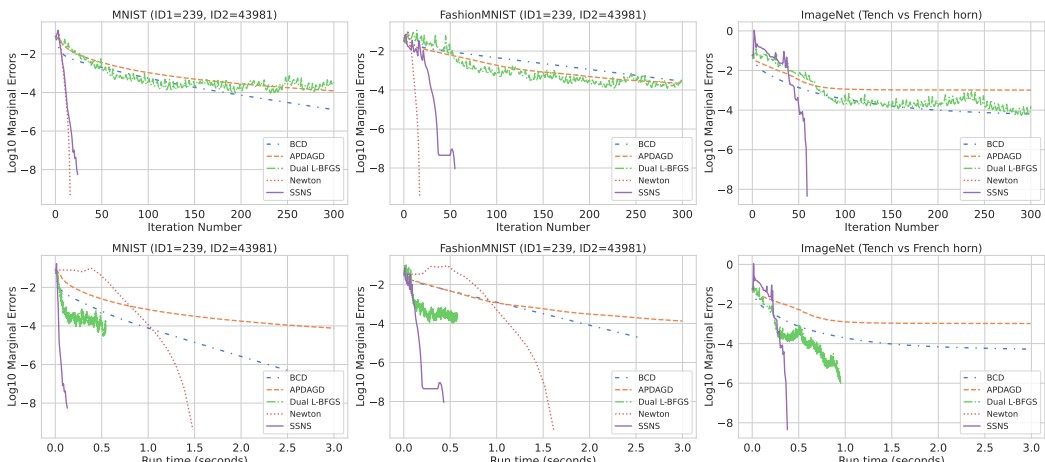

Figure 7: Additional test cases based on squared Euclidean distances and $\eta = 0.001$. Top: Marginal error vs. iteration number for different algorithms on three datasets. Bottom: Marginal error vs. run time.

so $D \geq 0$, $D_1 \mathbf{1}_{n+m-1} = \eta^{-1} \tilde{\Delta} \mathbf{1}_{m-1} \leq \eta^{-1} \delta \mathbf{1}_n$ and $D_2 \mathbf{1}_{n+m-1} = \eta^{-1} \tilde{\Delta}^T \mathbf{1}_n \leq \eta^{-1} \delta \mathbf{1}_{m-1}$. Overall, we have $D \mathbf{1}_{n+m-1} \leq \eta^{-1} \delta \mathbf{1}_{n+m-1}$. Since $D \geq 0$, this is equivalent to

$$\|D_{i\cdot}\|_1 \leq \eta^{-1} \delta, \quad i = 1, 2, \ldots, n+m-1.$$

Since $D$ is symmetric, we also have

$$\|D_{\cdot j}\|_1 = \|D_{j\cdot}\|_1 \leq \eta^{-1} \delta, \quad j = 1, 2, \ldots, n+m-1.$$

Let

$$P_k = \sum_{j \neq k} |D_{kj}|, \quad k = 1, \ldots, n+m-1,$$

and note that all diagonal elements of $D$ are zero. So by the Gershgorin circle theorem, every eigenvalue of $D$ must be smaller than or equal to the maximum of $D_{kk} + P_k = \|D_{k\cdot}\|_1$, which is upper bounded by $\eta^{-1} \delta$. Therefore, $\|D\| = \sigma_{\max}(D) \leq \eta^{-1} \delta$.

## B.2 Proof of Theorem (2)

Consider the matrix

$$A = \begin{bmatrix} \mathbf{diag}(r) & \tilde{T}_\delta \\ \tilde{T}_\delta^T & \mathbf{diag}(\tilde{c}) \end{bmatrix} \in \mathbb{R}^{(n+m-1)\times(n+m-1)},$$

and suppose that $T = (T_{ij})$, $T_\delta = (S_{ij})$. Clearly, $S_{ij} = T_{ij} - \Delta_{ij} \leq T_{ij}$. Let

$$P_k = \sum_{j \neq k} |A_{kj}|, \quad k = 1, \ldots, n+m-1,$$

and then it is easy to find that

$$A_{kk} + P_k = \begin{cases} r_k + \sum_{j=1}^{m-1} S_{kj} \leq 2r_k, & k = 1, \ldots, n \\ c_{k-n} + \sum_{i=1}^{n} S_{i,k-n} \leq 2c_{k-n}, & k = n+1, \ldots, n+m-1 \end{cases}.$$

Overall, we have $A_{kk} + P_k \leq U := 2\max\{\|r\|_\infty, \|c\|_\infty\}$. By the Gershgorin circle theorem, every eigenvalue of $A$ must be smaller than or equal to the maximum value of $A_{kk} + P_k$, which is upper bounded by $U < \infty$.

For some columns of $\Delta$, say, column $k$, it may be the case that $\Delta_{\cdot k} = 0$, indicating that there is no thresholding on $T_{\cdot k}$. Note that if we simultaneously permute the columns of $T$ and $T_\delta$, then the eigenvalues of $A$ do not change. Therefore, without loss of generality we can assume that the first $d$ columns and the last column of $\Delta$ are exactly zero, $0 \leq d \leq m-1$ ($\Delta_{\cdot m} = 0$ always holds by the design of Algorithm 1). Then we can partition $T_\delta$ as $T_\delta = [T^{(1)}, T_\delta^{(2)}, T_{\cdot m}]$, where $T^{(1)} \in \mathbb{R}^{n\times d}$ and $T_{\cdot m} \in \mathbb{R}^n$ have strictly positive elements. Accordingly, $A$ can be partitioned as

$$A = \begin{bmatrix} \mathbf{diag}(T\mathbf{1}_m) & T^{(1)} & T_\delta^{(2)} \\ T^{(1)T} & \mathbf{diag}(T^{(1)T}\mathbf{1}_n) & O \\ T_\delta^{(2)T} & O & \mathbf{diag}(T_\delta^{(2)T}\mathbf{1}_n) \end{bmatrix}.$$

Consider the matrix $B = A - svv^T$, where $v \in \mathbb{R}^{n+m-1}$ is a vector whose first $(n+d)$ elements are ones and the remaining elements are zeros, and $s$ is a positive scalar. Define

$$R_k = \sum_{j \neq k} |B_{kj}|, \quad k = 1, \ldots, n+m-1,$$

and suppose that $s \leq \min_{1 \leq i \leq n, 1 \leq j \leq d} T_{ij}$. Then for $k = 1, \ldots, n$, it is easy to find that

$$R_k = (n-1)s + \sum_{j=1}^{d}(S_{kj} - s) + \sum_{j=d+1}^{m-1} S_{kj} = (n-d-1)s + \sum_{j=1}^{m-1} S_{kj}.$$

For $k = n+1, \ldots, n+d$,

$$R_k = \sum_{i=1}^{n}(T_{i,k-n} - s) + (d-1)s = \sum_{i=1}^{n} T_{i,k-n} - (n-d+1)s.$$

For $k = n+d+1, \ldots, n+m-1$,

$$R_k = \sum_{i=1}^{n} S_{i,k-n} = \sum_{i=1}^{n} T_{i,k-n} - \sum_{i=1}^{n} \Delta_{i,k-n} < \sum_{i=1}^{n} T_{i,k-n}. \tag{7}$$

The inequality sign in (7) is strict, since we have assumed that at least one element of $\Delta_{\cdot,k-n}$ is nonzero. Moreover,

$$B_{kk} = \begin{cases} \sum_{j=1}^{m} T_{kj} - s, & k = 1, \ldots, n \\ \sum_{i=1}^{n} T_{i,k-n} - s, & k = n+1, \ldots, n+d \\ \sum_{i=1}^{n} T_{i,k-n}, & k = n+d+1, \ldots, n+m-1 \end{cases},$$

so overall we can obtain that

$$B_{kk} - R_k = \begin{cases} \sum_{j=1}^{m-1} \Delta_{kj} + T_{km} - (n-d)s, & k = 1, \ldots, n \\ (n-d)s, & k = n+1, \ldots, n+d \\ \sum_{i=1}^{n} \Delta_{i,k-n}, & k = n+d+1, \ldots, n+m-1 \end{cases}.$$

Now take

$$s = \frac{\min_{1 \le i \le n, 1 \le j \le m} T_{ij}}{2n},$$

and then we can verify that $s$ satisfies the condition $s \le \min_{1 \le i \le n, 1 \le j \le d} T_{ij}$. Moreover, for $k = 1, \ldots, n$, we have

$$B_{kk} - R_k \ge T_{km} - ns = T_{km} - \frac{1}{2} \cdot \min_{i,j} T_{ij} \ge \frac{1}{2} \cdot \min_{i,j} T_{ij}.$$

For $k = n+1, \ldots, n+d$,

$$B_{kk} - R_k = (n-d)s \ge (n-m+1)s = \frac{n-m+1}{2n} \cdot \min_{i,j} T_{ij}.$$

For $k = n+d+1, \ldots, n+m-1$, since $\Delta_{i,k-n}$ is either zero or $T_{i,k-n}$, and at least one element of $\Delta_{\cdot,k-n}$ is nonzero, we then have

$$B_{kk} - R_k = \sum_{i=1}^{n} \Delta_{i,k-n} \ge \min_{1 \le i \le n} T_{i,k-n} \ge \min_{i,j} T_{ij}.$$

Combining all the possibilities, we get

$$B_{kk} - R_k \ge L := \frac{n-m+1}{2n} \cdot \min_{i,j} T_{ij} > 0$$

for all $k$ and $d$.

By the Gershgorin circle theorem, every eigenvalue of $B$ must be greater than or equal to the minimum value of $B_{kk} - R_k$, which is lower bounded by $L > 0$. Since $A = B + svv^T$ and $vv^T$ is positive semi-definite, we have $\sigma_{\min}(A) \ge L > 0$, implying that $A$ is positive definite.

### B.3   Proof of Theorem 3

We first introduce some necessary notation. Define the level set as $L(x_0) = \{x : f(x) \le f(x_0)\}$, and let

$$\begin{aligned} f^* &= \inf\{f(x) : x \in L(x_0)\}, \\ \beta_1 &= \sup\{\|g(x)\| : x \in L(x_0)\}, \\ \beta_2 &= \sup\{\|H_\delta(x)\| : x \in L(x_0), \delta > 0\}. \end{aligned}$$

Define the function $\Phi(x, d) = f(x + d) - f(x) - g(x)^T d$. We then present a few technical lemmas, followed by the proof of the main theorem.

**Lemma 6.** $L(x_0)$ is a bounded and closed convex set, $f^* > -\infty$, $\beta_1 < \infty$, and $\beta_2 < \infty$.

*Proof.* Clearly, $f(x)$ is a continuously differentiable and strictly convex function. Define $L_c = \{x : f(x) \le c\}$, and then $L_c$ is a closed convex set.

Lemma 1 of Dvurechensky et al. [10] shows that if $(\alpha^*, \beta^*)$ is an optimal solution to (3), then

$$\max_j \beta_j^* - \min_j \beta_j^* \le R,$$

where $R = \|M\|_\infty / \eta - \log(\min_{i,j}\{a_i, b_j\})$. Since we have fixed $\beta_m^* = 0$, we immediately obtain that $\|\beta^*\| < \infty$. Given $\beta^*$, $\alpha^*$ has a closed form,

$$\alpha_i^* = \eta \log a_i - \eta \log \left[ \sum_{j=1}^{m} e^{\eta^{-1}(\beta_j^* - M_{ij})} \right],$$

so $\|\alpha^*\|$ is also bounded.

Let $x^* = (\alpha^{*T}, \tilde{\beta}^{*T})^T$ and $f^* = f(x^*)$, and then clearly $f^* > -\infty$. Take $c = c^* = \{f^*\}$, and obviously, $L_{c^*}$ is non-empty and bounded. Then Corollary 8.7.1 of [31] shows that $L_c$ is bounded for every $c$, implying that $L(x_0) = L_{f(x_0)}$ is bounded. Since $g(x)$ is continuous on $L(x_0)$ and $L(x_0)$ is compact, it is easy to show that $\beta_1 < \infty$.

By Theorem 2, $\|H_\delta\| = \sigma_{\max}(H_\delta) \leq 2\eta^{-1} \cdot \max\{\|r\|_\infty, \|c\|_\infty\}$, where $r = T\mathbf{1}_m$ and $c = T^T\mathbf{1}_n$. Clearly, every element of $T$ is bounded if $x$ is bounded, making $\|r\|_\infty$ and $\|c\|_\infty$ also bounded. As a result, $\|H_\delta\|$ is bounded on $L(x_0)$, implying that $\beta_2 < \infty$. $\qquad\square$

**Lemma 7.** *For any $\varepsilon > 0$, there exists a constant $\delta_0 > 0$ such that for all $\eta \in [c_l, c_u]$, $x \in L(x_0)$, and $\|p\| \leq \delta_0$ satisfying $x + \eta p \in L(x_0)$, we have*

$$|\Phi(x, \eta p)| \leq \frac{\rho_0 c_l \varepsilon}{4} \|p\|.$$

*Proof.* Since $f$ is continuously differentiable on $L(x_0)$, by the mean value theorem for multivariate functions, there exists some $\psi \in (0, 1)$ such that for all $x \in L(x_0)$ and $x + \eta p \in L(x_0)$,

$$f(x + \eta p) - f(x) = g(x + \psi \eta p)^T (\eta p).$$

Therefore,

$$\Phi(x, \eta p) = g(x + \psi \eta p)^T (\eta p) - g(x)^T (\eta p) = [g(x + \psi \eta p) - g(x)]^T (\eta p).$$

By Lemma 6, $g$ is continuous on $L(x^0)$ and $L(x^0)$ is a compact set, so $g$ is uniformly continuous on $L(x^0)$. Therefore, for any $\varepsilon > 0$, there exists a constant $\delta_0 > 0$ such that whenever $\|p\| \leq \delta_0$, we have

$$\|g(x + \psi \eta p) - g(x)\| \leq \frac{\rho_0 c_l \varepsilon}{4 c_u}.$$

As a result,

$$|\Phi(x, \eta p)| \leq \|g(x + \psi \eta p) - g(x)\| \cdot \|\eta p\| \leq \frac{\rho_0 c_l \varepsilon}{4} \|p\|.$$

$\qquad\square$

**Lemma 8.** *Let $\{x_k\}$ be generated by Algorithm 2, and then*

$$m_k(0) - m_k(\xi_k p_k) \geq \frac{1}{2} \|g_k\| \cdot \min\left\{ \|\xi_k p_k\|, \frac{\|g_k\|}{\|H_{\delta_k}\|} \right\}.$$

*Proof.* Let $f_k = f(x_k)$ and $r_k = \|\xi_k p_k\|$. If we take $(p, \lambda) = (\xi_k p_k, \mu_k \|g_k\|)$, and then we can verify that $(p, \lambda)$ meets the following relations:

$$\begin{cases} g_k + H_{\delta_k} p + \lambda p = 0 \\ \qquad\qquad\quad \lambda \geq 0 \\ \qquad\quad r_k - \|p\| \geq 0 \\ \quad \lambda \cdot (r_k - \|p\|) = 0. \end{cases}$$

As $m_k(\cdot)$ is a convex function, we have that $\xi_k p_k$ is a KKT point and an optimal solution to the constrained optimization problem

$$\min_p \quad m_k(p) = f_k + g_k^T p + \frac{1}{2} p^T H_{\delta_k} p$$
$$\text{s.t.} \quad \|p\| \leq r_k. \tag{8}$$

Next, consider the Cauchy point defined in Chapter 4 of Nocedal and Wright [24],

$$p_c^k = -\tau_k \frac{r_k}{\|g_k\|} g_k,$$

where

$$\tau_k = \begin{cases} 1, & \text{if } g_k^T H_{\delta_k} g_k \leq 0 \\ \min\left\{1, \frac{\|g_k\|^3}{r_k g_k^T V_k g_k}\right\}, & \text{if } g_k^T H_{\delta_k} g_k > 0 \end{cases}.$$

Then we can obtain the following inequality by Lemma 4.3 of Nocedal and Wright [24]:

$$m_k(0) - m_k(p_c^k) \geq \frac{1}{2}\|g_k\| \cdot \min\left\{r_k, \frac{\|g_k\|}{\|H_{\delta_k}\|}\right\}.$$

Meanwhile, the Cauchy point $p_c^k$ is a feasible point for the constrained problem (8), and $\xi_k p_k$ is an optimal solution for this problem. Therefore, $m_k(p_c^k) \geq m_k(\xi_k p_k)$, and we have

$$m_k(0) - m_k(\xi_k p_k) \geq \frac{1}{2}\|g_k\| \cdot \min\left\{r_k, \frac{\|g_k\|}{\|H_{\delta_k}\|}\right\}.$$

$\square$

**Lemma 9.** *Let $g_k$ and $\mu_k$ be generated by Algorithm 2. If there exists a constant $\varepsilon > 0$ and an integer $K$ such that $\|g_k\| \geq \varepsilon$ for all $k \geq K$, then there must exist a sufficiently large constant $\bar{\mu} > 0$, such that $\mu_{k+1} \leq 4\bar{\mu}$ for all $k \geq K$.*

*Proof.* For a given $\bar{\mu}$, define $I_1 = \{k : k \geq K, \mu_k < \bar{\mu}\}$ and $I_2 = \{k : k \geq K, \mu_k \geq \bar{\mu}\}$. If for some $\bar{\mu}$, $I_2$ is finite, then it trivially holds that all $\mu_k$ has a global upper bound, which leads to the desired conclusion. Therefore, we only consider the case that $I_2$ is infinite for all $\bar{\mu} > 0$. Since $\bar{\mu}$ can be chosen arbitrarily, $\{\mu_k\}_{k \in I_2}$ must be unbounded.

Then we will estimate the bound of the following quantity,

$$|\rho_k - 1| = \left|\frac{m_k(\xi_k p_k) - f(x_k + \xi_k p_k)}{m_k(0) - m_k(\xi_k p_k)}\right|.$$

According to Lemma 6, one has $\|g_k\| \geq \varepsilon$ and $\|g_k\| \leq \beta_1$ for $k \geq K$. Theorem 2 shows that $H_{\delta_k}$ is positive definite, so

$$\sigma_{\min}(H_{\delta_k} + \mu_k\|g_k\|) \geq \mu_k\varepsilon,$$
$$\left\|(H_{\delta_k} + \mu_k\|g_k\|I)^{-1}\right\| \leq \mu_k^{-1}\varepsilon^{-1}.$$

By Algorithm 2, we have

$$\xi_k p_k = \xi_k(H_{\delta_k} + \mu_k\|g_k\|I)^{-1}g_k,$$

and hence for all $k \in I_2$,

$$\|\xi_k p_k\| \leq \xi_k \left\|(H_{\delta_k} + \mu_k\|g_k\|I)^{-1}\right\| \cdot \|g_k\| \leq \xi_k\mu_k^{-1}\varepsilon^{-1}\beta_1. \tag{9}$$

Equation (9) indicates that $\|\xi_k p_k\|$, $k \in I_2$ can be made arbitrarily small with a sufficiently large $\bar{\mu}$. Therefore, we can choose some $\bar{\mu}$ such that

$$\|\xi_k p_k\| \leq \min\left\{\delta_0, \frac{\varepsilon}{\beta_2}, \frac{\rho_0 c_l \varepsilon}{2c_u\beta_2}\right\}, \quad \forall k \in I_2, \tag{10}$$

where $\delta_0$ is defined in Lemma 7. Meanwhile, we know

$$|m_k(\xi_k p_k) - f(x_k + \xi_k p_k)| = \left|\Phi(x_k, \xi_k p_k) - \frac{1}{2}(\xi_k p_k)^T H_{\delta_k}(\xi_k p_k)\right|$$

$$\leq |\Phi(x_k, \xi_k p_k)| + \frac{1}{2}\|H_{\delta_k}(\xi_k p_k)\| \cdot \|\xi_k p_k\|$$

$$\leq |\Phi(x_k, \xi_k p_k)| + \frac{c_u}{2}\|H_{\delta_k}\| \cdot \|\xi_k p_k\| \cdot \|p_k\|.$$

Lemma 7 indicates that when $\|\xi_k p_k\| \leq \delta_0$, we have

$$|\Phi(x_k, \xi_k p_k)| \leq \frac{\rho_0 c_l \varepsilon}{4}\|p_k\|.$$

Therefore, with $p_k$ that satisfies (10), we obtain

$$|m_k(\xi_k p_k) - f(x_k + \xi_k p_k)| \leq \frac{\rho_0 c_l \varepsilon}{4}\|p_k\| + \frac{c_u}{2} \cdot \beta_2 \cdot \frac{\rho_0 c_l \varepsilon}{2c_u\beta_2} \cdot \|p_k\| = \frac{\rho_0 c_l \varepsilon}{2}\|p_k\|.$$

On the other hand, Lemma 8 indicates that

$$m_k(0) - m_k(\xi_k p_k) \geq \frac{1}{2}\|g_k\| \cdot \min\left\{\|\xi_k p_k\|, \frac{\|g_k\|}{\|H_{\delta_k}\|}\right\}$$

$$\geq \frac{1}{2}\varepsilon \cdot \min\left\{\|\xi_k p_k\|, \frac{\varepsilon}{\beta_2}\right\}$$

$$= \frac{1}{2}\varepsilon \cdot \|\xi_k p_k\| \geq \frac{c_l \varepsilon}{2} \cdot \|p_k\|,$$

so we have

$$|\rho_k - 1| = \left|\frac{m_k(\xi_k p_k) - f(x_k + \xi_k p_k)}{m_k(0) - m_k(\xi_k p_k)}\right| \leq \frac{\frac{\rho_0 c_l \varepsilon}{2}\|p_k\|}{\frac{c_l \varepsilon}{2} \cdot \|p_k\|} = \rho_0,$$

which means that $\rho_k \geq 1 - \rho_0$ for all $k \in I_2$. Then by the design of Algorithm 2, we have $\mu_{k+1} \leq \mu_k$ for all $k \in I_2$.

Now we can show that $\mu_{k+1} \leq 4\bar{\mu}$ for all $k \geq K$ by induction. First, enlarge $\bar{\mu}$ when necessary to ensure that $\bar{\mu} > \mu_K$. Then by Algorithm 2, we must have $\mu_{K+1} \leq 4\bar{\mu}$. Now suppose that $\mu_{l+1} \leq 4\bar{\mu}$ for some $l \geq K$. If $\mu_{l+1} \in I_1$, then clearly $\mu_{l+2} \leq 4\bar{\mu}$ immediately holds. Otherwise, $\mu_{l+1} \in I_2$, so by the argument above, we have $\mu_{l+2} \leq \mu_{l+1} \leq 4\bar{\mu}$. In both cases, the conclusion holds. □

**Lemma 10.** *Under the same conditions as in Lemma 9, define*

$$\mathcal{K} = \{k : k \geq K, \rho_k \geq \rho_0\}.$$

*Then $\mathcal{K}$ must be a finite set.*

*Proof.* We use proof by contradiction. Suppose that $\mathcal{K}$ is an infinite set. Since

$$\rho_k = \frac{f(x_k) - f(x_k + \xi_k p_k)}{m_k(0) - m_k(\xi_k p_k)},$$

and note that $x_{k+1} = x_k$ if $\rho_k < 0$, we have $f(x_{k+1}) \leq f(x_k + \xi_k p_k)$. Therefore, by Lemma 8, it holds that

$$f(x_k) - f(x_{k+1}) \geq \rho_0[m_k(0) - m_k(\xi_k p_k)] \geq \frac{\rho_0}{2}\|g_k\| \cdot \min\left\{\|\xi_k p_k\|, \frac{\|g_k\|}{\|H_{\delta_k}\|}\right\}$$

$$\geq \frac{\rho_0}{2}\varepsilon \cdot \min\left\{\|\xi_k p_k\|, \frac{\varepsilon}{\beta_2}\right\}$$

for all $k \in \mathcal{K}$. Combined with Lemma 6, we find that $f(x_k)$ is monotonically non-increasing with a lower bound, so $f(x_k)$ has a limit, and hence

$$\lim_{k \in \mathcal{K}, k \to \infty} \|\xi_k p_k\| = 0.$$

On the other hand, $p_k = -(H_{\delta_k} + \mu_k\|g_k\|I)^{-1}g_k$, which means that $g_k = -(H_{\delta_k} + \mu_k\|g_k\|)p_k$. Therefore,

$$\varepsilon \leq \|g_k\| = \|(H_{\delta_k} + \mu_k\|g_k\|I)p_k\| \leq \|H_{\delta_k} + \mu_k\|g_k\|\| \cdot \|p_k\|$$

$$\leq [\|H_{\delta_k}\| + \mu_k\|g_k\|] \cdot \|p_k\| \leq (\beta_2 + \mu_k\beta_1) \cdot \frac{1}{c_l} \cdot \|\xi_k p_k\|.$$

In other words,

$$0 \leq \frac{c_l \varepsilon}{\beta_2 + \mu_k\beta_1} \leq \|\xi_k p_k\| \to 0,$$

which implies that $\mu_k \to \infty$ for $k \in \mathcal{K}$, $k \to \infty$. This contradicts with the fact that $\mu_k \leq 4\bar{\mu}$ as shown in Lemma 9. So to conclude, $\mathcal{K}$ must be a finite set. □

**Lemma 11.** *Let $\{x_k\}$ be generated by Algorithm 2. Then either Algorithm 2 terminates in finite iterations, or $g(x_k)$ satisfies*

$$\liminf_{k \to \infty} \|g(x_k)\| = 0.$$

*Proof.* We prove this lemma by contradiction. Suppose that there exist some $\varepsilon > 0$ and an integer $K$ such that

$$\|g_k\| > \varepsilon, \quad \forall k \geq K.$$

Then Lemma 10 implies that the index set

$$\mathcal{K} = \{k : k \geq K, \rho_k \geq \rho_0\}$$

is infinite. This means that there is a sufficiently large integer $K'$ such that $\rho_k < \rho_0$ for all $k \geq K'$. According to Algorithm 2, we must have

$$\mu_{k+1} = 4\mu_k, \quad \forall k \geq K',$$

which means that $\mu_k \to \infty$. However, this contradicts with the fact that $\mu_k \leq 4\bar{\mu}$ for some $\bar{\mu} > 0$ as shown in Lemma 9. Therefore, we must have

$$\liminf_{k \to \infty} \|g_k\| = 0.$$

$\square$

Then we are ready to prove Theorem 3. Let $x^*$ be the unique global optimum of (4). Clearly, $f(x)$ and $g(x)$ are Lipschitz continuous on $L(x_0)$, so there exist constants $C_1, C_2 > 0$ such that

$$|f(x_k) - f(x^*)| \leq C_1 \|x_k - x^*\|, \tag{11}$$
$$\|g(x_k) - g(x^*)\| \leq C_2 \|x_k - x^*\|. \tag{12}$$

On the other hand, by taking $\delta = 0$, Theorem 2 shows that there is a constant $c_1 > 0$ such that $\sigma_{\min}(H(x)) \geq c_1$ for all $x \in L(x_0)$, so there is a constant $c_2 > 0$ such that

$$\|g(x_k)\| = \|g(x_k) - g(x^*)\| \geq c_2 \|x_k - x^*\|. \tag{13}$$

Also by Taylor's theorem,

$$f(x_k) = f(x^*) + [g(x^*)]^T (x_k - x^*) + \frac{1}{2}(x_k - x^*)^T H(z_1)(x_k - x^*),$$

where $z$ is some point between $x_k$ and $x^*$. Since $g(x^*) = 0$, we obtain

$$f(x_k) - f(x^*) \geq \frac{c_1}{2}\|x_k - x^*\|^2. \tag{14}$$

By the design of Algorithm 2, $f(x_k)$ is non-increasing and is lower bounded by $f^* = f(x^*)$, so $f(x_k)$ must have a limit. Suppose that

$$\lim_{k \to \infty} |f(x_k) - f^*| = \varepsilon \geq 0,$$

and then (11) and (13) indicate that for sufficiently large $k$,

$$\frac{1}{2}C_1^{-1}\varepsilon \leq C_1^{-1}|f(x_k) - f(x^*)| \leq \|x_k - x^*\| \leq c_2^{-1}\|g(x_k)\|.$$

Since in Lemma (11) we have shown that $\liminf_{k \to \infty} \|g_k\| = 0$, $\varepsilon$ cannot be any positive value. Therefore, $\varepsilon = 0$. Combining (12) and (14), we have

$$\frac{c_1}{2C_2^2}\|g_k\|^2 \leq \frac{c_1}{2}\|x_k - x^*\|^2 \leq f(x_k) - f(x^*) \to \varepsilon = 0,$$

which implies that $\lim_{k \to \infty} \|g_k\| = 0$ and $\lim_{k \to \infty} \|x_k - x^*\| = 0$.

### B.4 Proof of Theorem 4

For convenience, let $H_k = H(x_k)$ and $B_k = H_{\delta_k} + \mu_k \|g_k\| I$. Theorem 2 shows that for any $\delta$,

$$\sigma_{\min}(H_\delta) \geq \eta^{-1} \cdot \frac{n - m + 1}{2n} \cdot \min_{i,j} T_{ij}.$$

Note that

$$T = \tau(\alpha, \beta) = \left( e^{\eta^{-1}(\alpha_i + \beta_j - M_{ij})} \right),$$

and each $(\alpha_i + \beta_j)$ is bounded on $L(x_0)$, so each element of $T$ must be bounded below from zero on $L(x_0)$. Therefore, there exists a constant $0 < C_1 < \infty$ such that $\sigma_{\min}(H_\delta) \geq C_1^{-1}$ for all $x \in L(x_0)$.

By the design of Algorithm 2, $x_k \in L(x_0)$ for all $k$. As a result,

$$\|B_k^{-1}\| = [\sigma_{\min}(B_k)]^{-1} \leq [\sigma_{\min}(H_{\delta_k})]^{-1} \leq C_1, \tag{15}$$

and hence $\|p_k\| = \|B_k^{-1} g_k\| \leq C_1 \|g_k\|$ for all $k$. Moreover, Theorem 2 shows that $\|H_{\delta_k}\| = \sigma_{\max}(H_{\delta_k}) \leq C_2$ for some $C_2 > 0$, so by Lemma 8, we have

$$
\begin{aligned}
m_k(0) - m_k(\xi_k p_k) &\geq \frac{1}{2} \|g_k\| \cdot \min \left\{ \|\xi_k p_k\|, \frac{\|g_k\|}{\|H_{\delta_k}\|} \right\} \\
&\geq \frac{1}{2C_1} \|p_k\| \cdot \min \left\{ c_l \|p_k\|, \frac{\|p_k\|}{C_1 C_2} \right\} \\
&\geq \frac{\min\{c_l, C_1^{-1} C_2^{-1}\}}{2C_1} \cdot \|p_k\|^2 := C_3 \|p_k\|^2.
\end{aligned}
$$

On the other hand,

$$
\begin{aligned}
|m_k(\xi_k p_k) - f(x_k + \xi_k p_k)| &= \left| f(x_k + \xi_k p_k) - f(x_k) - g_k^T(\xi_k p_k) - \frac{1}{2}(\xi_k p_k)^T H_{\delta_k}(\xi_k p_k)^T \right| \\
&\leq \left| f(x_k + \xi_k p_k) - f(x_k) - g_k^T(\xi_k p_k) - \frac{1}{2}(\xi_k p_k)^T H_k(\xi_k p_k)^T \right| \\
&\quad + \left| \frac{1}{2}(\xi_k p_k)^T (H_k - H_{\delta_k})(\xi_k p_k)^T \right|. \tag{16}
\end{aligned}
$$

By Taylor's theorem,

$$\left| f(x_k + \xi_k p_k) - f(x_k) - g_k^T(\xi_k p_k) - \frac{1}{2}(\xi_k p_k)^T H_k(\xi_k p_k)^T \right| = o(\|\xi_k p_k\|^2).$$

For the second term of (16), let $D_k = H_k - H_{\delta_k}$, and then Theorem 1 shows that $\|D_k\| \leq \eta^{-1} \delta_k$. By Algorithm 2, $\delta_k \leq \nu_0 \|g_k\|^\gamma$, so $\|H_k - H_{\delta_k}\| \leq \eta^{-1} \nu_0 \|g_k\|^\gamma$. Then

$$\left| \frac{1}{2}(\xi_k p_k)^T (H_k - H_{\delta_k})(\xi_k p_k)^T \right| \leq \frac{\nu_0}{2\eta} \|g_k\|^\gamma \cdot \|\xi_k p_k\|^2 = o(\|\xi_k p_k\|^2).$$

As a result, $|m_k(\xi_k p_k) - f(x_k + \xi_k p_k)| = o(\|\xi_k p_k\|^2)$, and hence

$$|\rho_k - 1| = \left| \frac{m_k(\xi_k p_k) - f(x_k + \xi_k p_k)}{m_k(0) - m_k(\xi_k p_k)} \right| \leq \frac{o(\|\xi_k p_k\|^2)}{C_3 \|p_k\|^2} \to 0.$$

This implies that there is an integer $K > 0$ such that for all $k \geq K$, $\rho_k \geq 1 - \rho_0$. By the design of Algorithm (2), we have $\mu_{k+1} \leq \kappa$ and $x_{k+1} = x_k + \xi_k p_k$ for all $k \geq K$.

### B.5 Proof of Theorem 5

We first present a classical result derived from the mean value theorem of vector-valued functions.

**Lemma 12** (Theorem 3.2.12 of [26]). *Let $g : D \subset \mathbb{R}^n \to \mathbb{R}^m$ be continuously differentiable on a convex set $D_0 \subset D$ and suppose that for constants $\alpha \geq 0$ and $p \geq 0$, $\nabla g$ satisfies*

$$\|\nabla g(u) - \nabla g(v)\| \leq \alpha \|u - v\|^p, \quad \forall u, v \in D_0.$$

*Then, for any $x, y \in D_0$,*

$$\|g(y) - g(x) - \nabla g(x)(y - x)\| \leq [\alpha/(p+1)] \cdot \|y - x\|^{p+1}. \tag{17}$$

Then we are ready to prove Theorem 5. Let $x^*$ be the unique global optimum of (4), and then $g(x^*) = 0$. Clearly, $g(x)$ and $H(x)$ are Lipschitz continuous on $L(x_0)$, so there exist constants $L_1, L_2 > 0$ such that for all $x, y \in L(x_0)$,

$$
\begin{aligned}
\|g(x) - g(y)\| &\leq L_1 \|x - y\|, \\
\|H(x) - H(y)\| &\leq L_2 \|x - y\|.
\end{aligned}
$$

This implies that $g$ satisfies the conditions in Lemma 12 with $\alpha = L_2$ and $p = 1$. By substituting $x \leftarrow x_k$ and $y \leftarrow x^*$, we have

$$\|g(x_k)\| = \|g(x_k) - g(x^*)\| \leq L_1 \|x_k - x^*\|,$$

$$\|g(x^*) - g(x_k) - H(x_k)(x^* - x_k)\| = \|g(x_k) - g(x^*) - H(x_k)(x_k - x^*)\| \leq \frac{1}{2} L_2 \|x_k - x^*\|^2.$$

Without loss of generality let $K' > K$, and then for all $k \geq K'$, $\mu_k \leq \kappa$ and $x_{k+1} = x_k + p_k$. In the proof of Theorem 5, we have shown that

$$\|H_k - H_{\delta_k}\| \leq \eta^{-1} \nu_0 \|g_k\|^\gamma,$$

so for $\|g_k\| \leq 1$,

$$\|H_k - B_k\| \leq \|H_k - H_{\delta_k}\| + \mu_k \|g_k\| \leq (\eta^{-1} \nu_0 + \kappa) \|g_k\| \leq L_1 (\eta^{-1} \nu_0 + \kappa) \|x_k - x^*\|.$$

Also note that $\|B_k^{-1}\| \leq C_1$ from (15), and then we have

$$\begin{aligned}
\|x_{k+1} - x^*\| = \|x_k + p_k - x^*\| &= \|x_k - x^* - B_k^{-1} g_k\| \\
&= \|B_k^{-1}(B_k(x_k - x^*) - g_k)\| \\
&\leq \|B_k^{-1}\| \cdot \|g_k - g(x^*) - H_k(x_k - x^*) + H_k(x_k - x^*) - B_k(x_k - x^*)\| \\
&\leq C_1 \left(\|g_k - g(x^*) - H_k(x_k - x^*)\| + \|(H_k - B_k)(x_k - x^*)\|\right) \\
&\leq C_1 \left(\frac{1}{2} L_2 \|x_k - x^*\|^2 + \|H_k - B_k\| \cdot \|x_k - x^*\|\right) \\
&\leq C_1 \left(\frac{1}{2} L_2 \|x_k - x^*\|^2 + L_1 (\eta^{-1} \nu_0 + \kappa) \|x_k - x^*\|^2\right) \\
&= C_1 \left(\frac{1}{2} L_2 + L_1 (\eta^{-1} \nu_0 + \kappa)\right) \|x_k - x^*\|^2.
\end{aligned}$$

