# OpenReview forum: "Safe and Sparse Newton Method for Entropic-Regularized Optimal Transport"
_NeurIPS.cc/2024/Conference — NeurIPS 2024 poster_

### Official Review · Reviewer_bgcB · 2024-07-10

**Soundness:** 3
**Presentation:** 3
**Contribution:** 3
**Rating:** 6
**Confidence:** 3

**Summary:**

This paper proposed a new Newton-type algorithm for entropic-regularized optimal transport (OT) which utlizes sparsification and safeguard techniques and achieves global convergence and local quadratic convergence for the entropic-OT problem. Numerical experiments are provided to verify the effectiveness of the proposed method.

**Strengths:**

The paper is well-rounded and well-presented. The paper addresses the computational issue caused by a dense Hessian by sparsification, and addresses the singularity issue by a safeguard mechanism. Both the local and global convergences are provided. Numerical experiments are provided to show the effectiveness of the proposed method.

**Weaknesses:**

(Please answer the Questions section directly) The safeguard mechanism in this paper is actually a standard trust region technique; The numerical experiment settings could be improved.

**Questions:**

In general, I think this is a technically solid paper. The authors identify the issue of applying second order methods to entropic-OT and use sparsification and safeguard to make it applicable. I didn't check the detail of the proof, but believe it to be correct since these all follow the standard techniques. I'd recommend this work to be accepted.

I have the following comments and questions:

1. Regarding Theorem 2 which studies the positive-definiteness of the sparsified matrix $H_{\delta}$: what is the largest and smallest eignvalues if we don't conduct Algorithm 1, meaning we use the original Hessian in (5) directly? The author should add some discussion on this to consolidate their point of "safely used to compute the Newton search directions".

2. When the authors talk about "safe", does it mean Theorem 2 or the trust region type update on line 7-13 in Algorithm 2, or both?

3. In the experiments for MNIST and ImageNet, the authors should provide the dimension information corresponds to the theory part, such as what $n$ and $m$ are.

4. In all images, Newton method seems to be very slow. Is this all due to the costly process of solving the dense linear system?

5. What is $\tilde{T}$ in (5)? I feel that it's not clearly defined. Is it corresponds to $\tilde{\beta}$ which is $T$ having one row/column removed?

**Limitations:**

The authors are clear about their limitations.

---

> ### Author Rebuttal · Authors · 2024-08-07
>
> ### **Weaknesses**
>
> Thanks for the comments. First of all, we shall clarify that SSNS is directly inspired by the regularized Newton method [R6] and the Levenberg–Marquardt algorithm [R7,R8], which have close relations to the trust-region methods but with some subtle differences. Specifically, standard trust-region methods first set a trust-region radius $\Delta_k$ in each iteration, and then (approximately) solve a subproblem to determine the search direction. In SSNS, we update the shift parameter $\lambda_k$ instead, and the search direction has a closed form. The formula of search directions in trust-region methods has a similar form, but its shift $\lambda_k$ is implicitly determined by $\Delta_k$, and typically requires some root-finding techniques. Also, SSNS allows for a step size selection procedure as in Algorithm 3, thus enhancing the flexibility.
>
> Also per the suggestions of reviewers, we have improved the numerical experiments as explained in the global rebuttal.
>
> ### **Questions**
>
> 1. Recall that in Theorem 2 we allow $\delta=0$, which corresponds to the case that we use the genuine Hessian matrix. In this setting, Algorithm 2 reduces to a conventional regularized Newton method, but its per-iteration cost would be huge. Theorem 2 is an improvement to existing sparsifed Newton methods such as [R3], since previous works do not guarantee that the sparsified Hessian is invertible (but note that the true Hessian matrix is always invertible by Theorem 2).
> 2. In our original design of the algorithm, "safe" mainly refers to the positive definiteness of the sparsified Hessian matrix. But from the experiments, we also find that the shift parameter (which has a close relation to trust-region techniques) is very useful, as the classical Newton method fails in some examples due to numerical instability.
> 3. Thanks for the suggestion. We have added such information in our local revision. Basically it is $n=m=784$ for (Fashion-)MNIST and $n\approx m\approx 1000$ for ImageNet data.
> 4. Yes. This can be inferred from the fact that the number of iterations of the Newton method is very small, but its runtime is large. The huge per-iteration cost is the consequence of a dense linear system.
> 5. Yes. We have defined this notation in the beginning of Section 2, meaning removing the last column of $A$.
>
> [R3] Tang, X., Shavlovsky, M., Rahmanian, H., Tardini, E., Thekumparampil, K. K., Xiao, T., & Ying, L. (2024). Accelerating Sinkhorn algorithm with sparse Newton iterations.
>
> [R6] Li, D. H., Fukushima, M., Qi, L., & Yamashita, N. (2004). Regularized Newton methods for convex minimization problems with singular solutions.
>
> [R7] Levenberg, K. (1944). A method for the solution of certain non-linear problems in least squares.
>
> [R8] Marquardt, D. W. (1963). An algorithm for least-squares estimation of nonlinear parameters.

---

> > ### Comment · Reviewer_bgcB · 2024-08-08
> >
> > I thank the authors for their rebuttal. I decide to keep my score as I think this is a good work and the authors do clarify most of my concerns.

---

> > > ### Author Response · Authors · 2024-08-09
> > >
> > > Thanks for the feedback!

---

### Official Review · Reviewer_gzRs · 2024-07-12

**Soundness:** 3
**Presentation:** 3
**Contribution:** 3
**Rating:** 5
**Confidence:** 3

**Summary:**

This paper proposes to solve the entropy-regularized OT problem by proposing a customized Newton method. More specifically, the authors propose a new way to sparsify the Hessian matrix as well as adaptive mechanisms to adjust the hyper-parameters in each iteration. Experimental results are also encouraging.

**Strengths:**

1. The new sparsification method is novel and different from [31]. I like the fact that you can prove that this method positive definite matrices.

2. The self-adaptive mechanism to adjust Newton method hyper-parameters look interesting. I am not 100% positive (considering that there are already line search methods to guarantee convergence of Newton methods), but this part seems novel to me. The authors only cite very old papers [16, 17, 21] as motivations.

**Weaknesses:**

1. The major part is the scalability of the proposed method. Newton-variants methods are second-order methods. The largest scale the authors have tried is on the transformed features of ImageNet. The feature dimension is only 30. If we use the Sinkhorn algorithm, I believe we can handle much large scales.

2. The experimental results are only sample results based on very few samples from the dataset. I am concerned that the results could be cherry-picked. It would be much more comprehensive if the authors can present training efficiency improvement on the overall dataset.

3. There is no experimental comparison with SNS [31] even though this work takes a step further on top of SNS. Looking at the SNS paper, I believe it is very easy to implement.

**Questions:**

1. What is the computational complexity of Algorithm 1 Sparsification?
2. What is the computational complexity of each iteration in Algorithm 2?
3. Can you compare with the SNS method and report results in tables during rebuttal?
4. The SNS paper uses the metric Log of Optimality Gap while you use the metric Log of Marginal Errors? What is the difference? Can you report your results in terms of Log of Optimality Gap as well during rebuttal?
5. Can you also replicate the Table 3 and Table 4 in the SNS paper? Basically conduct perturbation study on the entropy regularization parameters.
6. For the imageNet dataset, how well does your proposed method perform if you change the final feature dimension from 30 to 60 and 90, respectively?

---

> ### Author Rebuttal · Authors · 2024-08-07
>
> Thanks for the comments. Below are our point-by-point responses.
>
> ### **Weaknesses**
>
> 1. The dimension of the features does not impact the scale the problem. The extracted features are only used to compute the cost matrix, which has a size of $n\times m$. We do not make the feature dimension $d$ too large mainly because it is commonly accepted that the Euclidean distance suffers from the curse of dimensionality. In the global rebuttal, we have added experiments to study the impact of feature dimension and the scalability of algorithms on very large problems.
> 2. As we have mentioned in the global rebuttal and other threads, to enhance the reproducibility, we do not pick the pairs of images. Instead, the randomly selected image IDs are taken from the prior literature [R1] that studies quadratically regularized OT, and we follow their experiment setting to study entropic-regularized OT. We have included different examples in Figures 4 and 5 in Appendix A.3. Also, to investigate the scalability of algorithms on very large problems, we have added experiments to test the performance for $n=m=1000,5000,10000$.
> 3. We agree that the overall framework of SNS [R3] is easy to understand, but the major challenges for directly comparing with SNS are: (a) there is no publicly available code of SNS to reproduce the experiments; (b) in [R3], the algorithm has many hyperparameters, and we have no clear scheme to set those hyperparameters. For example, how many Sinkhorn iterations to run before switching to the Newton step, how large the sparsification parameter should be, what the fallback strategy should be when the linear system is not invertible, etc. In fact, one of the major motivations of this article is to make the SNS framework more adaptive and practical, and we position our SSNS algorithm as a concrete and practical variant of the original SNS method.
>
> ### **Questions**
>
> 1. A trivial upper bound of the computational complexity of Algorithm 1 is $O(n^2\log(n))$, assuming $n=m$. This is done by sorting the values of each column or row of $T$. However, if we have an estimated upper bound $\varrho$ of the density of the sparsified $T$, meaning that each column or row of $T$ has at most $\varrho n$ nonzero elements after sparsification, then the cost can be reduced to $O(n^2+\varrho n^2\log(\varrho n))$. This is done by selecting the largest $\varrho n$ elements in each column or row of $T$, and only sort values within these $\varrho n$ elements. In a typical setting, $\varrho$ is very small due to the approximate sparsity of $T$.
> 2. Each iteration of Algorithm 2 has a computational cost of $O(n^2)$ to compute the gradient and objective function value, plus the cost in Algorithm 1, plus the cost in solving the sparse linear system. When $T$ is very sparse, the number of nonzero elements of $T$ may be as small as $O(n)$, so the overall computation is dominated by the first and second parts, *i.e.*, $O(n^2)$.
> 3. As we have explained above, there are certain practical difficulties in directly comparing with SNS. But we have tried our best to provide more experiment results similar to those in the SNS paper, as introduced in the points below.
> 4. A typical definition of the optimality gap is the value $f(x_k)-f(x^*)$, where $f(\cdot)$ is the objective function, and $x^*$ is the optimal point. The marginal error in entropic-regularized OT coincides with the gradient norm $\Vert g(x_k) \Vert$. We do not use the optimality gap because the ground-truth optimal point $x^*$ is in general unknown, and needs to be approximated using existing solvers. But this process also introduces rounding errors, and may be very sensitive when $x_k$ is close to convergence. Instead, $\Vert g(x_k) \Vert$ is easy to compute, and has an absolute lower bound of zero. Overall, we think that the marginal error is a more objective and easy-to-compute criterion to evaluate convergence.
> 5. Yes, we have added such experiments in the global rebuttal.
> 6. Per the suggestion, we have added such experiments in the global rebuttal (Figure R2).
>
> [R1] Pasechnyuk, D. A., Persiianov, M., Dvurechensky, P., & Gasnikov, A. (2023). Algorithms for Euclidean-regularised optimal transport.
>
> [R3] Tang, X., Shavlovsky, M., Rahmanian, H., Tardini, E., Thekumparampil, K. K., Xiao, T., & Ying, L. (2024). Accelerating Sinkhorn algorithm with sparse newton iterations.

---

> > ### Comment · Reviewer_gzRs · 2024-08-14
> >
> > Thank you very much for your reply.
> >
> > Most of my questions have answered, except the requested comparison with [31], which is the following work:
> >
> >     [31] Tang, X., Shavlovsky, M., Rahmanian, H., Tardini, E., Thekumparampil, K. K., Xiao, T., & Ying, L. (2024). Accelerating Sinkhorn algorithm with sparse newton iterations.
> >
> > This might sound a little bit too harsh, but I have some leftover question on whether there are any author overlaps between this submission and [31].
> > If there is an author overlap, then you defintely have access to the code base in [31].
> > Some research groups do this so that they can publish another paper with impressive results, without having to beat against their previous method.
> > I apologize if this sounds too mean.
> >
> > However, I wouldn't know this because of NeurIPS's double blind policy. I will give you the benefit of doubt and trust you that there is no author overlap and you do not have access to the codebase of [31]. I do want to leave this comment here, just in case this possibility happens.
> >
> > As for now, I will maintain my score since it is already above the acceptance threshold.

---

> > > ### Author Response · Authors · 2024-08-14
> > >
> > > Dear reviewer,
> > >
> > > We really appreciate your comments and totally understand your concern. To avoid violating the double blind policy, we have communicated with the Area Chairs to handle this issue. What we can comment here is that we have tried our best to enhance the reproducibility and integrity of this work, and the comparison with [31] is subject to some practical difficulties. Given this situation, we have added similar experiments in [31] per your suggestions, which are contained in the global rebuttal.
> > >
> > > Thanks for your understanding.

---

### Official Review · Reviewer_DpgA · 2024-07-12

**Soundness:** 2
**Presentation:** 3
**Contribution:** 2
**Rating:** 5
**Confidence:** 4

**Summary:**

This paper proposes a Newton-based algorithm to solve the entropic optimal transport problem on the basis of samples. The approach hinges on a "sparsification" scheme for the Hessian (which is explained in Algorithm 1) that retains many favorable properties such as an approximation error due to the sparsification (Theorem 1) and positive definite-ness (Theorem 2). The conclude with numerical experiments on image-data, where they demonstrate that their algorithm significantly outperforms other algorithms for EOT on the examples they considered.

**Strengths:**

The paper is well-written (with very few typos), with the overall story and methodology quite clear.

**Weaknesses:**

The experiments are mildly underwhelming, and also a bit inconsistent. In the first set of experiments, the authors are running their new (E)OT algorithm between two images, where the pixel frequency denotes the weights for the EOT problem. In the second set of experiments, where they perform (E)OT between two classes of images, there they flatten the images, performing some pre-processing so they lie in d=30, and use $n\simeq m \simeq 1000$ samples to constitute the weight vectors ($1/n\bm{1}$ and $1/m\bm{1}$). These are relatively small-scale in the number of samples $n$, and I am inclined to believe that it does not scale well when $n=10000$ or more, which the block-coordinate-descent approach (i.e., the vanilla Sinkhorn algorithm) does allow for. I could be wrong, but this is the more realistic/modern use-case for EOT, which is not covered in the experiments.

**Questions:**

See weaknesses.

- Following the above, is it possible to provide a runtime in the style of Sinkhorn given a prescribed tolerance level?
- Can this approach be adapted for unbalanced entropic OT?

---

> ### Author Rebuttal · Authors · 2024-08-07
>
> ### **Weaknesses**
>
> Thanks for the comments. As we have explained in the global rebuttal and other threads, the experiments are intentionally designed to reflect two typical uses of OT, one for image morphing and interpolation, and the other for computing statistical distances. In our local revision, we have added a section in the appendix to explain the motivation of these experiments.
>
> As for the scalability, we have added a new set of experiments at the scales of $n=m=1000,5000,10000$, as explained in the global rebuttal. The results show that SSNS is capable of solving entropic-regularized OT on very large problems, and is more efficient than BCD in terms of run time and number of iterations.
>
> ### **Questions**
>
> 1. We suppose the "runtime" here refers to the theoretical computational complexity based on the global convergence speed. First of all, to the best of our knowledge, most convergence rate results for Newton-type methods are local, and there is few on the global convergence rate. However, there is a simple "ensemble" method to safeguard the global convergence speed. Note that our Algorithm 2 never increases the objective function value in any iteration, so we can append a Sinkhorn iteration after every $N$ Newton-type steps, where $N$ is a fixed integer. In this way, we obtain at least a Sinkhorn-like convergence speed, but are likely to make larger progresses during the Newton steps. This idea is also mentioned in Section 3.2, page 41 of [R4].
> 2. Following the formulation in [R5], entropic-regularized unbalanced OT also has a smooth dual objective function (equation (4) of [R5]). Its difference with the balanced version is that the linear term in equation (4) of our article,
> $$-\alpha^T a-\beta^T b=-\sum_{i=1}^n \alpha_i a_i-\sum_{j=1}^m \beta_j b_j,$$
> is replaced by sums of exponentials,
> $$\tau \sum_{i=1}^n e^{-\alpha_i/\tau} a_i+\tau \sum_{j=1}^m e^{-\beta_j/\tau} b_j.$$
> Clearly, this means that only the diagonal elements of the Hessian matrix would be different from the setting in our article, and we think that the proposed sparsification scheme and the Newton-type algorithm still apply.
>
> [R4] Nocedal, J., & Wright, S. J. (2006). Numerical optimization.
>
> [R5] Pham, K., Le, K., Ho, N., Pham, T., & Bui, H. (2020). On unbalanced optimal transport: An analysis of Sinkhorn algorithm.

---

> > ### Comment · Reviewer_DpgA · 2024-08-12
> >
> > Thanks for the comments. I will still keep my score as-is.

---

### Official Review · Reviewer_vqB6 · 2024-07-12

**Soundness:** 4
**Presentation:** 4
**Contribution:** 4
**Rating:** 8
**Confidence:** 3

**Summary:**

The paper proposes a Newton method to solve the entropic-regularized optimal transform (OT) problem.  The method includes a novel strategy for the sparse approximation of the Hessian to reduce computational complexity compared to the classical Newton method and applying a diagonal shift on the sparse Hessian to avoid singularity. The sparse approximation of the Hessian is a simple procedure based on removing the small off-diagonal block elements;  the authors have theoretically proven that the approximation error is bounded and that the sparse approximate is always positive definite. The authors have demonstrated the efficacy of their method on two OT problems.

**Strengths:**

1.	The paper is very well-written, organized and clear. I congratulate the authors for writing a paper that is this easy to read while they make significant theoretical contributions.
2.	The proposed method is practical with only few tuning parameters.
3.	The theoretical guarantees of the proposed method are impressive and this is important in eliminating the limitations of prior art.
4.	The numerical experiments clearly demonstrate the advantages of the proposed method in the error performance and computational complexity.

**Weaknesses:**

I have not found any major weaknesses. Minor comments/typos:
1.	The authors have not explained how they chose the pair of images for their first experiment. Is it random? Does the algorithm have similar performance for other images?
2.	The authors have usen $K$ for both the number of small elements and the number of iterations. Please use a different symbol for either one to avoid confusion.
3.	Line 67: “proved” should be “proven.”
4.	Line 110: The full name of “BFGS” is missing.
5.	What the authors mean by “safe” only becomes clear at line 169. I would suggest making this clarification earlier in the paper. Maybe, Contribution #2 could be rephrased so that is clear that “safe” refers to avoiding singularity.

**Questions:**

I find the first experiment setting a bit confusing. Could the authors give the motivation behind this experiment? Why are the pixel values used as probabilities? Please see Weaknesses for some other questions on this experiment.

**Limitations:**

The authors have mentioned the limitation of their work.

---

> ### Author Rebuttal · Authors · 2024-08-07
>
> Thank you for the comments. Below are our point-by-point responses for the questions.
>
> ### **Weaknesses**
>
> 1. To enhance the reproducibility, we do not pick the pairs of images. Instead, the randomly selected image IDs are taken from the prior literature [R1] that studies quadratically regularized OT, and we follow their experiment setting to study entropic-regularized OT. In fact, we have already included other image pairs in Figures 4 and 5 in Appendix A.3, and the IDs of image pairs can be seen from the titles of images. In the rebuttal PDF we have added more test cases in Figure R1.
>
> 2-4. Thanks for the suggestions. We have fixed these issues in our local revision.
>
> 5. Thanks for the suggestion. We call our proposed algorithm a safe and sparse Newton method, in the sense that the linear systems for computing the search directions are always positive definite. This property addresses the invertibility issues of existing sparsified Newton methods, and is crucial for practical implementation. In our local manuscript, we have added explanations in the introduction, above the contribution section.
>
> ### **Questions**
>
> We are trying to demonstrate two typical uses of OT. In the first experiment, images are vectorized as density vectors, and OT is a used as a tool for image morphing and interpolation [R2]. The experiment setting is already used in previous literature such as [R1] and [R3]. The second experiment uses OT as a statistical distance to measure the difference between two distributions. In this setting, each image is one observation of a distribution, and we use OT to compute the (approximate) Wasserstein distance between two classes of images. In our local revision, we have added a section in the appendix to explain the motivation of these experiments.
>
> [R1] Pasechnyuk, D. A., Persiianov, M., Dvurechensky, P., & Gasnikov, A. (2023). Algorithms for Euclidean-regularised optimal transport.
>
> [R2] Papadakis, N. (2015). Optimal transport for image processing.
>
> [R3] Tang, X., Shavlovsky, M., Rahmanian, H., Tardini, E., Thekumparampil, K. K., Xiao, T., & Ying, L. (2024). Accelerating Sinkhorn algorithm with sparse newton iterations.

---

> > ### Comment · Reviewer_vqB6 · 2024-08-08
> >
> > Thank you for the clarifications!

---

### Author Rebuttal · Authors · 2024-08-07

# To All Reviewers

Thank you all reviewers for the encouraging and insightful comments. We appreciate the time and effort the reviewers have dedicated to providing valuable feedback on our manuscript. In this round, we have made every effort to address the comments of the reviewers. **The point-by-point responses are provided in the reply**.

Additionally, we would like to take this opportunity to make some global clarifications as well as improvements we have made during this period.

### **Explaining what "safe" means**

We call our proposed algorithm a safe and sparse Newton method, in the sense that the linear systems for computing the search directions are always positive definite. This property addresses the invertibility issues of existing sparsified Newton methods, and is crucial for practical implementation.

### **Motivations on the experiment setting**

We are trying to demonstrate two typical uses of OT when designing the numerical experiments. In the first experiment, images are vectorized as density vectors, and OT is a used as a tool for image morphing and interpolation [R2]. The experiment setting is already used in previous literature such as [R1] and [R3]. The second experiment uses OT as a statistical distance to measure the difference between two distributions. In this setting, each image is one observation of a distribution, and we use OT to compute the (approximate) Wasserstein distance between two classes of images.

### **Improvements on experiments**

Per the suggestions of reviewers, we have improved the numerical experiments on the following aspects:

1. We make it clear that to enhance **the reproducibility**, we do not pick the pairs of images in the numerical experiments. Instead, the randomly selected image IDs are taken from the prior literature [R1] that studies quadratically regularized OT, and we follow their experiment setting to study entropic-regularized OT. In the article, we have included other image pairs in Figures 4 and 5, and in the rebuttal PDF we have added more test cases in Figure R1.
2. We have studied **the impact of feature dimension** $d$ in the ImageNet experiment (Figure R2, rebuttal PDF). The plots imply that the convergence property of SSNS is robust to the feature dimension of input images.
3. We have studied **the impact of regularization parameters** on the performance of optimization algorithms. The table below shows the performance comparison between BCD and SSNS under different regularization parameters for the ImageNet experiment in Section 5. The convergence tolerance is set to $\varepsilon_{tol}=10^{-8}$, and the cost matrix is based on the $\ell_{1}$-distance.

|$\log_{10}(\eta)$|BCD Time (s)|BCD Iterations|SSNS Time (s)|SSNS Iterations|
|-|-|-|-|-|
|-2   | 1.628| 217|1.523|13 |
|-2.25|>3.765|>500|0.960|20 |
|-2.5 |>3.765|>500|0.461|30 |
|-2.75|>3.766|>500|0.383|57 |
|-3   |>3.767|>500|0.771|120|

The table below shows the case using cost matrices based on the squared Euclidean distances.

|$\log_{10}(\eta)$|BCD Time (s)|BCD Iterations|SSNS Time (s)|SSNS Iterations|
|-|-|-|-|-|
|-2   |0.438 |59  |2.235|11|
|-2.25|0.853 |114 |1.066|15|
|-2.5 |3.327 |443 |0.997|23|
|-2.75|>3.773|>500|0.529|35|
|-3   |>3.773|>500|0.458|68|

The results show that BCD is very sensitive to the value of $\eta$. When $\eta$ is large, BCD may demonstrate some computational advantages, but when $\eta$ is small, BCD typically fails to meet the error tolerance within 500 iterations. The pattern of SSNS shows some interesting points: when $\eta$ becomes smaller, the number of iterations also increases, but the overall runtime of SSNS may even decrease. This is because smaller $\eta$ values typically result in more sparse Hessian approximations, thus leading to faster sparse linear system solving. These findings are consistent with our explanations in Section 5.

4. We have investigated **the scalability** of SSNS on very large OT problems. We consider a synthetic OT problem that can generate data with arbitrary dimensions. The basic setting is to approximate the OT between two continuous distributions: the source is an exponential distribution with mean one, and the target is a normal mixture distribution $0.2\cdot N(1,0.2)+0.8\cdot N(3,0.5)$. We discretize the problem in the following way: let $x_i=5(i-1)/(n-1)$, $i=1,\ldots,n$, and $y_j=5(j-1)/(m-1)$, $j=1,\ldots,m$, which are equally-spaced points on [0, 5]. Define the cost matrix as $M_{ij}=(x_i-y_j)^{2}$. Let $f_1$ and $f_2$ be the density functions of the source and target distributions, respectively. Then we set $\tilde{a}_i=f_1(x_i)$, $\tilde{b}_j=f_2(y_j)$, $a\_i=\tilde{a}\_i/\left(\sum\_{k=1}^n\tilde{a}\_k\right)$, and $b\_j=\tilde{b}\_j/\left(\sum\_{k=1}^m\tilde{b}\_k\right)$. Similar to the experiment setting in Section 5, we normalize the cost matrix and set $\eta=0.001$. We then solve the entropic-regularized OT problem using BCD and SSNS at the scales of $n=m=1000,5000$, and 10000. The results are visualized in Figure R3 in the rebuttal PDF, whose pattern is clear: BCD demonstrates a linear-like convergence rate, and SSNS has a fast convergence speed consistent with the theoretical quadratic rate. Thanks to the Hessian sparsification, SSNS does not suffer from a high per-iteration cost, so overall it provides an efficient solver for entropic-regularized OT even on very large problems.

[R1] Pasechnyuk, D. A., Persiianov, M., Dvurechensky, P., & Gasnikov, A. (2023). Algorithms for Euclidean-regularised optimal transport.

[R2] Papadakis, N. (2015). Optimal transport for image processing.

[R3] Tang, X., Shavlovsky, M., Rahmanian, H., Tardini, E., Thekumparampil, K. K., Xiao, T., & Ying, L. (2024). Accelerating Sinkhorn algorithm with sparse newton iterations.

---

### Decision · Program_Chairs · 2024-09-25

**Decision:**

Accept (poster)

**Comment:**

From the reviews, the reviewers reach a consensus that the paper contains interesting result and novel contribution to the community. The Author-Reviewer discussions were also substantial, the authors should incorporate the important ones to the final version of the paper or supplementary to futher strengthen the paper. During the Reviewer-AC discussions, there are two points raised regarding the numerical experiments, if time allows, it would be great if the authors conduct the experiments
 - From Reviewer DpgA: in the comparison against BCD, what the result would be if the tolerance is relatively larger (still acceptable)? As for this setting, BCD would be much faster.
 - From Reviewer gzRs: For showing the scalability of the method, it is applied to solve OT on large-scale synthetic datasets. However, on the real-world datasets such as imageNet (Figure R2 in the rebuttal pdf), the authors have only conducted on dimension of 30, 60, and 90. For revision, it would be great if the authors could solve OT on the imageNet for dimension of 500 and 1000, respectively. If they can show that the same computational advantage can hold on real-world datasets.